# NanoBaseLib: A Multi-Task Benchmark Dataset for Nanopore Sequencing

**Guangzhao Cheng**[1]      **Chengbo Fu**[1]      **Lu Cheng**[1,2,*]

[1] Department of Computer Science, Aalto University, Finland
[2] Institute of Biomedicine, University of Eastern Finland, Finland

## Abstract

Nanopore sequencing is the third-generation sequencing technology with capabilities of generating long-read sequences and directly measuring modifications on DNA/RNA molecules, which makes it ideal for biological applications such as human Telomere-to-Telomere (T2T) genome assembly, Ebola virus surveillance and COVID-19 mRNA vaccine development. However, accuracies of computational methods in various tasks of Nanopore sequencing data analysis are far from satisfactory. For instance, the base calling accuracy of Nanopore RNA sequencing is ∼90%, while the aim is ∼99.9%. This highlights an urgent need of contributions from the machine learning community. A bottleneck that prevents machine learning researchers from entering this field is the lack of a large integrated benchmark dataset. To this end, we present NanoBaseLib, a comprehensive multi-task benchmark dataset. It integrates 16 public datasets with over 30 million reads for four critical tasks in Nanopore data analysis. To facilitate method development, we have preprocessed all the raw data using a uniform workflow, stored all the intermediate results in uniform formats, analysed test datasets with various baseline methods for four benchmark tasks, and developed a software package to easily access these results. NanoBaseLib is available at `https://nanobaselib.github.io`.

## 1 Introduction

**Biological background.** Nanopore sequencing represents the third-generation sequencing technology characterized by its ability to produce ultra-long reads ($>10^4$ bp). In contrast, second-generation sequencing, commonly known as next-generation sequencing (NGS) exemplified by Illumina sequencing, can only generate short reads (<300 bp) [1]. Oxford Nanopore Technologies (ONT) is the main provider of Nanopore sequencing products. Due to its unique technological advantages, Nanopore sequencing has been used in various biological applications. The well-known applications include Telomere-to-Telomere (T2T) human genome project [2], rapid sequencing of SARS-CoV-2 genomes [3], Ebola virus surveillance [4], and direct RNA sequencing [5] that facilitates COVID-19 mRNA vaccine developments [6].

Nanopore sequencing technology measures the current signal by translocating a DNA/RNA molecule through a nanoscale pore anchored on a lipid membrane, as illustrated in Figure 1. The shape, size, and chemical properties of nucleotides in the pore jointly determine the current signals, which are collected and utilized to infer the nucleotide sequence using computational methods (e.g., seq2seq deep learning models). There are multiple nucleotides (usually 5) in the pore, which we term as $k$mer (e.g., 5mer). The inferred $k$mers are assembled into a DNA/RNA sequence computationally, which completes the sequencing process. Compared to Illumina sequencing, Nanopore sequencing not only

---

*Correspondence to: lu.cheng.ac@gmail.com.

38th Conference on Neural Information Processing Systems (NeurIPS 2024) Track on Datasets and Benchmarks.

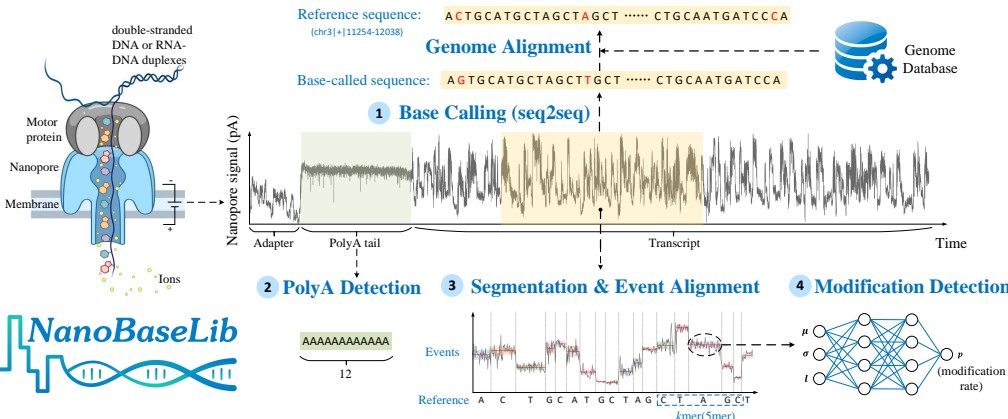

Figure 1: Nanopore sequencing and data analysis. NanoBaseLib consists of four benchmark tasks: (1) **B**ase **C**alling (**BC**), (2) **P**olyA **D**etection (**PD**), (3) **S**egmentation and Event **A**lignment (**SA**), and (4) **M**odification **D**etection (**MD**).

facilitates the sequencing of ultra-long molecules but also preserves the epi-transcriptomic information on DNA/RNA molecules. However, its lower base-calling accuracy poses a significant limitation, impacting the performance of various downstream tasks (see further discussion in Appendix Methods 4.1).

**Lack of large integrated benchmark dataset.** We want to draw the attention of the machine learning community to the prevailing challenges within the field of Nanopore sequencing. However, there are several barriers for machine learning researchers to work in this field. First, the full training data of ONT company is proprietary, which is not accessible to the public research community. Second, public Nanopore sequencing data are scattered in different repositories, each of which is generally only utilized for one computational task. They could be used for other computational tasks if preprocessed into a suitable form. Third, raw Nanopore sequencing data is large (typical file size is ∼200GB), which takes a long time to preprocess. Fourth, dependencies exist between different tasks. If one aims to focus on a downstream task, it is necessary to execute all upstream tasks to obtain the preprocessed inputs for the downstream task. This process demands extensive bioinformatic expertise and is time-consuming. Lastly, there are diverse applications of Nanopore sequencing, which use different library preparation protocols and bioinformatic preprocessing workflows. To alleviate the barriers for a general machine learning researcher, we try to (1) identify the key computational tasks, (2) collect a large training dataset, and (3) develop a unified benchmark framework for different computational tasks. In this work, we primarily focus on transcriptomics (mRNA) related applications, which is one of the major applications of Nanopore sequencing.

The general transcriptome analysis workflow for Nanopore sequencing consists of four major tasks (as illustrated in Figure 1): base calling, polyA detection, segmentation and event alignment, and base modification detection. **Base calling** is the fundamental step that converts a chunk of raw current signals into a DNA/RNA sequence. This task typically employs a seq2seq model, which is analogous to automatic speech recognition (ASR) systems. Subsequent to base calling, the DNA/RNA sequence is mapped to the reference genome, which provides its genomic location and the corresponding transcript. **PolyA detection** serves as a quality control measure after base calling. Since the polyA tail is used in the sequencing library preparation, and any sequence without a detectable polyA tail is flagged as a low-quality read and eliminated from downstream analysis. In the **segmentation and event alignment** task, the raw current signal is segmented into small fragments and aligned with the $k$mers of the reference genome sequence. An aligned $k$mer that corresponds to a current signal segment is termed as an "event". The **modification detection** task identifies whether a base is modified. Typically, this is determined by a specific event that corresponds to a particular modification state of the associated $k$mer. For example, it assesses whether the central nucleotide "A" in the $k$mer "GGACT" is an unmodified "A" or a modified "m6A" ($N^6$-methyladenosine). Generally, the input of each task depends on the output of the previous task. An overview of these tasks (input, output and machine learning methods) is provided in Appendix Table 3. In order to develop better models for these tasks, building a large training dataset and a multi-task benchmark framework is essential.

However, there exist various challenges, which we will discuss from three perspectives: dataset, benchmark, and ground truth acquisition.

**Challenges in building the dataset and benchmark.** From the dataset collection perspective, the primary challenge is managing the substantial volume of the raw data (TB scale) in Nanopore sequencing, which is stored in binary data formats such as fast5 and pod5. Another challenge is that ONT has released various versions of the pores so far, ranging from R6 to the latest R10.3 [7]. Each version requires a separate set of parameters in preprocessing. Further, it takes a significant amount of time and effort to find relevant datasets from the literature, explore the datasets, figure out the technical details and preprocess the data, e.g., some papers only provide the basecalled sequences rather than the raw data.

In order to develop a fair benchmark, we need to set a unified standard in the data preprocessing. The bioinformatic preprocessing of collected datasets is complex and demanding as different datasets may be generated using different pore versions, from different species, adopted different ground truth, etc. These datasets are preprocessed using different workflows in the original publications, e.g., basecalled using different software, mapped to different genome versions, which undermines the fairness if directly used for benchmarking purposes. Therefore, we carefully selected the bioinformatic tools and configured the correct parameters to develop a unified preprocessing pipeline, which is used to re-process all datasets from the raw data.

Furthermore, obtaining the ground truth for various tasks requires lots of domain knowledge. For example, the ground truth for base calling involves mapping raw current signal to the reference genome. After that the matched raw signal and reference sequence need to be further sampled to improve the generalization capabilities of neural networks. The ground truth of RNA modification requires the analysis of various ChIP-seq data.

**Our solution.** In this work, we present NanoBaseLib, a comprehensive dataset and benchmarking framework designed to address the above challenges. Our main contributions are as follows:

- We build a comprehensive Nanopore sequencing dataset by integrating 16 public datasets with over 30 million reads, all preprocessed using the unified pipeline. The dataset can be used for various tasks and includes a range of RNA modifications (m6A, m5C, hm5C, inosine, pseudouridine).

- We benchmark four key tasks of Nanopore sequencing based on the unified, preprocessed dataset. These tasks include base calling, polyA detection, segmentation and event alignment, and RNA modification detection.

- We develop a software package that streamlines the incorporation of new datasets, enabling efficient preprocessing and integration of newly available Nanopore data.

- We deploy a user-friendly website to facilitate easier navigation of datasets, tasks, and benchmark results.

In summary, we wish NanoBaseLib could help machine learning researchers to solve the challenges in the Nanopore sequencing field and contribute to answering bigger biological questions.

## 2  Related Work

Nanopore sequencing technologies are rapidly advancing, leading to a surge in related publications. We focus on key papers and resources from two perspectives: datasets and benchmarks. Datasets emphasize data resources, while benchmarks concentrate on comparative analyses.

**Databases.** ONT has created the Oxford Nanopore Open Data project [8], which provides exemplar datasets for different applications of Nanopore sequencing. These datasets are generally task-specific, which is not ideal for training large networks. Jain et al. [9] provide a large number of Nanopore DNA sequencing samples (n=53) for genome assembly of a human cell line. This dataset can only be utilized for the Nanopore DNA base calling task and is limited to one species. DirectRMDB [10] is a database covering 16 types of RNA modifications and 25 species inferred from Nanopore RNA sequencing data using existing tools. Due to the lack of intermediate results in uniform formats, this database is not very helpful for developing new methods.

**Benchmarks.** There are several Nanopore base calling benchmark studies [11–13], which are limited to one species and only focus on one task. In contrast, Magi et al. [14] covers multiple tasks such as base calling, de novo assembly, and variant discovery, but their analysis is limited to DNA data. Zhong et al. [15] benchmark ten methods on the m6A detection task from RNA sequencing data. However, it does not cover the base calling, polyA detection and segmentation & alignment tasks. As will be discussed in Section 3.2, researchers need to make a significant effort to re-process the raw data and obtain these intermediate results.

In summary, there is currently a lack of a comprehensive benchmark platform that integrates multi-task, multi-species Nanopore DNA/RNA datasets, along with corresponding intermediate results and a user-friendly software package to facilitate the utilization of these results.

## 3 Datasets

### 3.1 Data collection

The Nanopore raw signal file is stored in fast5 or pod5 format, which contains both the raw signal and various meta information, as illustrated in Appendix Figure 2. The raw signal refers to the recorded current signals as a DNA/RNA molecule translocates through a pore. The meta information provides other related information such as the sequencing kit version, protocol version, flowcell version, channel information, etc. The meta information is important for raw data processing. For example, ONT uses different pores (proteins) in different products, thus the raw signal needs to be adjusted for different pores. A single flow cell of the sequencing device can contain 512 to 2,675 pores, each of which is termed as a channel. There is a set of channel specific parameters to convert the raw signal of a given channel to the picoampere (pA) scale. The typical size of raw fast5 or pod5 files in a single dataset is approximately 200 GB. The fast5 or pod5 file and the reference genome sequence are the basic input files for our analyses.

By exploring the literature, we collect Nanopore sequencing datasets by the following criteria. First, the raw data (fast5 or pod5) could be freely downloaded from a public repository. Second, we try to incorporate datasets from diverse species, e.g. *Homo sapiens*, *Mus musculus*, *Escherichia Coli* etc. Third, NanoBaseLib tries to cover multiple common RNA modifications such as m6A, m5C, inosine and pseudouridine. Fourth, the availability of ground truth for multiple tasks is a major consideration in the development. Table 1 presents an overview of the collected datasets from public repositories [9, 12, 16–22], comprising 16 datasets with over 30 million reads from six species, including both raw and preprocessed data. All these datasets we collected are based on R9.4 Nanopore chemistry. Comprehensive details and statistics of the dataset are provided in Appendix Table 1 and Table 2.

### 3.2 Preprocessing pipeline

The inputs of different tasks rely on preprocessing of the raw data. To ensure a fair comparison in the benchmark, a unified standard for data preprocessing is adopted in NanoBaseLib. Since the downloaded public datasets are preprocessed using different pipelines with different software, it is challenging to reuse their preprocessed results. We directly start from the raw data and preprocess them using the same pipeline, whose outputs are used as the inputs for different tasks. The pipeline consists of six steps, each of which could be implemented using different software packages. We choose several methods for each step by following common practices, which also serve as the baseline methods in different tasks. The overview and limitations of all software and tools used in the pipeline are summarized in Appendix Table 4. The unified pipeline is depicted in Appendix Figure 1 and can be summarized as follows:

**Step 1: Raw data format standardization.** Raw data are typically available in three formats: single-fast5, multi-fast5, and pod5. For standardization, all datasets are converted into the multi-fast5 format using the `ont_fast5_api` tool from the pod5 package.

**Step 2: Base calling.** This step converts raw current signals into DNA/RNA sequences. The raw current signal file is base called using Guppy, which generates a FASTQ file consisting of many reads, i.e. DNA/RNA sequences. Detailed software information is provided in Appendix Table 4, and the configuration parameters are provided in Appendix Table 5.

Table 1: Datasets overview of NanoBaseLib.

| Dataset | Raw Data Size (GB) | Species | Type | Task BC[1] | PD[2] | SA[3] | MD[4] |
|---|---|---|---|---|---|---|---|
| ont_polya_standard | 81 | *Synthetic* | RNA | ✓ | ✓ | ✓ | ✗ |
| eGFP_polyA_DNA | 43 | *Synthetic* | cDNA | ✓ | ✓ | ✓ | ✗ |
| eGFP_polyA_RNA | 529 | *Synthetic* | RNA | ✓ | ✓ | ✓ | ✗ |
| lambda_phage | 19 | *Lambda phage* | DNA | ✓ | ✗ | ✓ | ✗ |
| NA12878 | 68 | *Homo sapiens* | DNA | ✓ | ✗ | ✓ | ✗ |
| curlcake | 584 | *Synthetic* | RNA | ✓ | ✗ | ✓ | ✓ |
| scBY4741_m5C | 37 | *Synthetic* | RNA | ✓ | ✗ | ✓ | ✓ |
| scBY4741_hm5C | 17 | *Synthetic* | RNA | ✓ | ✗ | ✓ | ✓ |
| scBY4741_pU | 4 | *Synthetic* | RNA | ✓ | ✗ | ✓ | ✓ |
| hct116 | 346 | *Homo sapiens* | RNA | ✓ | ✗ | ✓ | ✓ |
| hek293t_wt | 224 | *Homo sapiens* | RNA | ✓ | ✗ | ✓ | ✓ |
| hek293t_ko | 356 | *Homo sapiens* | RNA | ✓ | ✗ | ✓ | ✗ |
| mESCs_eligos | 220 | *Mus musculus* | RNA | ✓ | ✗ | ✓ | ✓ |
| ecoli_eligos | 214 | *Escherichia coli* | RNA | ✓ | ✗ | ✓ | ✓ |
| dinopore_ivt | 15 | *Synthetic* | RNA | ✓ | ✗ | ✓ | ✓ |
| dinopore_xenopus | 399 | *Xenopus lavies* | RNA | ✓ | ✗ | ✓ | ✓ |

[1] Base calling, [2] PolyA detection, [3] Segmentation and event alignment, [4] Modification detection.

**Step 3: Mapping to the reference genome.** This step maps the base called reads to the reference genome. The mapping is performed using minimap2 [23] with parameter `-ax map-ont`.

**Step 4 (optional): PolyA detection (only for RNA).** This step detects the polyA signal for each read in the raw data. Since polyA tail is utilized in the sequencing library preparation, a read without a polyA tail should be removed. Thus this step serves as a quality control step. NanoBaseLib provides two kinds of polyA detection results using Nanopolish `polya` [24] and Tailfindr [17]. This step will output the polyA detection flag, start and end positions of the polyA segment on the raw signal, polyA length (number of "A" nucleotides).

**Step 5: Segmentation and event alignment.** This step aligns the raw signal segments with the DNA/RNA sequences. The DNA/RNA sequence is converted into a list of consecutive $k$mers. Given the mapping results (from step 3) and the raw current file, Nanopolish `eventalign`, Tombo `resquiggle` and SegPore `eventalign` [25] will output the segmentation and event alignment files. Here the segmentation refers to a segment of raw signals and the event refers to a $k$mer (e.g. GGACT) corresponding to a nucleotide (e.g. A) at a certain genomic location (see Appendix Figure 6).

**Step 6 (optional): Modification detection.** This step estimates the RNA modification probability of a nucleotide (or $k$mer) at a given genomic location. Note that we only estimate one type of RNA modification (e.g. m6A, m5C, pseudouridine) at a time, i.e. a method generally estimates only one type of RNA modification.

In Nanopore sequencing, one of the important factors causing batch effect [26] is the sequencing bias generated by different sequencing devices. Our unified pipeline can mitigate batch effects to a certain extent. Firstly, each ONT sequencing device (e.g. MinION) will generate a set of device-specific parameters to convert the raw signal into pico Ampere (pA) values, ensuring consistency between different sequencing devices. Subsequently, different downstream tools will standardize or normalize the signal further. For example, Tombo employs median shift and median absolute deviation (MAD) to normalize the current signal [27], while Nanopolish uses scaling parameters to account for per-read variations [24]. These processes help to remove batch effects between different datasets to a certain extent. The signal conversion and normalization processes in our preprocessing pipeline are detailed in the Appendix Methods 4.2.

## 3.3 Data Storage

There are two types of data in NanoBaseLib: the raw data and the pre-processed data. The raw data come from the original publications and could be downloaded from the original repositories. Our

website also offers a summary of raw data download links for easier access (`https://nanobaselib.github.io/raw.html`). The pre-processed data and results following the unified pipeline can be downloaded from NanoBaseLib website (`https://nanobaselib.github.io/dataset.html`) or Zenodo (`https://doi.org/10.5281/zenodo.10889896`).

# 4 Benchmarks

We select four critical tasks (as illustrated in Figure 1) for benchmarking: **B**ase **C**alling (**BC**), **P**olyA **D**etection (**PD**), raw signal **S**egmentation and event **A**lignment (**SA**), and RNA **M**odification **D**etection (**MD**). BC converts the raw current signal into DNA/RNA sequences, which is the fundamental task. PA detects the polyA tail signal from each read, which mainly serves as a quality control. SA segments the raw current signal and aligns the derived segments with reference sequence, i.e. match a raw signal segment with a $k$mer in the reference sequence. SA is the essential task to provide interpretations to downstream tasks, as the raw signal segment carries the chemical information of the aligned $k$mer. MD classifies if a $k$mer (the central nucleotide) at a certain genomic location carries a certain RNA modification, based on the raw signal segments aligned to this $k$mer. MD is currently one of the major challenges of the RNA research community, which is crucial for downstream applications such as mRNA vaccine development. These four tasks are interdependent, and the input of the next task generally depends on the output of the previous task.

The test datasets used in the benchmarks were randomly selected, and none were utilized in the training of any baseline models. Detailed descriptions of the baselines used in the benchmarks are provided in Appendix Table 3 and on our NanoBaseLib website. In the following, we will discuss each task separately, focusing on four key aspects: ground truth acquisition, evaluation metrics, baseline models and benchmark performance.

## 4.1 Base Calling (BC) benchmark

**Ground truth acquisition.** The base calling task aims to convert the raw current signal into DNA/RNA bases. In this task, the ground truth is established through a one-to-one correspondence between raw signal segments and fragments of the reference genome sequence. To obtain this ground truth, we first perform base calling (using Guppy v6.0.1), then align the base sequence with the reference genome sequence and run Nanopolish `eventalign`. Finally, we extract the matched raw signal segments and reference sequence fragments as the ground truth. This process is illustrated in Appendix Figure 3. In total, the BC benchmark dataset comprises 34,742 (DNA) and 83,681 (RNA) pairs of raw signal chunks and their corresponding reference sequence segments.

**Evaluation metrics.** To assess the performance of various base calling algorithms, we conduct pairwise alignment between the predicted base called sequences and the ground truth sequences. This alignment process enables the quantification of similarity between the predicted and reference sequences. Based on the resulting pairwise alignments, we count matches ($M$), mismatches ($X$), insertions ($I$), and deletions ($D$) and derive the corresponding rates as follows:

$$\frac{M/X/I/D}{\text{\# base of alignment}} \times 100\% \qquad (1) \qquad\qquad \frac{M/X/I/D}{\text{\# base of reference}} \times 100\% \qquad (2)$$

which normalize the counts based on the length of the alignment (Eq. 1) or the length of ground truth sequence (Eq. 2). Higher match ($M$) rate indicates better base calling performance, while higher mismatch ($X$), insertion ($I$) or deletion ($D$) rates mean worse performance. These metrics are computed by applying the pairwise alignment algorithm implemented in the parasail package [28].

**Baseline models.** We select both official ONT basecallers and third-party basecallers for the benchmark. Guppy, Bonito, and Dorado are official basecallers from ONT, while the others (Causalcall [29], Rodan [30]) are developed by the research community. Multiple versions of Guppy and Dorado are benchmarked as they are the most commonly used basecallers. All methods are based on deep learning models. The software versions and their limitations are listed in Appendix Table 4.

**Benchmark performance.** As shown in Table 2 (DNA) and Table 3 (RNA), Guppy (v6.0.1) and Dorado (v0.5.3) achieved the best performance in DNA and RNA base calling, respectively. The best accuracy (match rate) is approximately 93%. Compared to the general accuracy of second-generation sequencing (99.9%), there is significant potential for improvement.

Table 2: DNA base calling performances on test dataset NA12878.

| Model | $\frac{M}{Align}$ ↑ | $\frac{I}{Align}$ ↓ | $\frac{X}{Align}$ ↓ | $\frac{D}{Align}$ ↓ | $\frac{M}{Ref}$ ↑ | $\frac{I}{Ref}$ ↓ | $\frac{X}{Ref}$ ↓ | $\frac{D}{Ref}$ ↓ |
|---|---|---|---|---|---|---|---|---|
| Causalcall | 84.30 | 0.82 | 4.11 | 10.77 | 84.41 | 0.82 | 4.12 | 10.79 |
| Guppy(v2.3.1) | 86.63 | 2.59 | 4.34 | 6.44 | 88.08 | 2.64 | 4.36 | 6.53 |
| Guppy(v4.5.4) | 91.93 | 1.97 | 2.68 | 3.42 | 93.27 | 2.01 | 2.71 | 3.48 |
| Guppy(v6.0.1) | **93.60** | **1.51** | **2.12** | **2.77** | **94.30** | **1.54** | **2.14** | **2.79** |
| Bonito(v0.7.3) | 93.35 | 1.56 | 2.23 | 2.86 | 94.13 | 1.59 | 2.25 | 2.90 |
| Dorado(v0.5.3) | 93.35 | 1.57 | 2.24 | 2.85 | 93.07 | 1.57 | 2.23 | 2.84 |
| Dorado(v0.7.0) | 93.47 | 1.54 | 2.18 | 2.81 | 93.17 | 1.55 | 2.17 | 2.80 |

Table 3: RNA base calling performances on test dataset hek293t_wt.

| Model | $\frac{M}{Align}$ ↑ | $\frac{I}{Align}$ ↓ | $\frac{X}{Align}$ ↓ | $\frac{D}{Align}$ ↓ | $\frac{M}{Ref}$ ↑ | $\frac{I}{Ref}$ ↓ | $\frac{X}{Ref}$ ↓ | $\frac{D}{Ref}$ ↓ |
|---|---|---|---|---|---|---|---|---|
| Rodan | 87.72 | 3.28 | 4.91 | 4.08 | 85.16 | 3.05 | 4.17 | 3.78 |
| Guppy(v2.3.1) | 85.67 | 3.59 | 4.46 | 6.29 | 88.06 | 3.74 | 4.54 | 6.43 |
| Guppy(v4.5.4) | 91.78 | 2.39 | 2.15 | 3.67 | 93.40 | 2.48 | 2.19 | 3.74 |
| Guppy(v6.0.1) | 91.78 | 2.39 | 2.15 | 3.67 | 93.40 | 2.48 | 2.19 | 3.74 |
| Dorado(v0.5.3) | **93.96** | **1.89** | **1.81** | **2.34** | **95.22** | **1.96** | **1.84** | **2.37** |
| Dorado(v0.7.0) | 93.74 | 1.95 | 1.92 | 2.40 | 95.01 | 2.02 | 1.94 | 2.43 |

## 4.2 PolyA Detection (PD) benchmark

The PD task aims to identify the raw signal segment of the polyA tail and outputs the length of "A" bases in the polyA tail. Specifically, the PD task outputs the polyA detection flag, start and end positions of the polyA segment on the raw signal and the polyA length (number of "A" nucleotides). However, only the polyA length has ground truth, while the others are not available due to technological limitations.

**Ground truth acquisition.** Three synthetic datasets (`ont_polya_standard`, `eGFP_polyA_DNA`, and `eGFP_polyA_RNA`) used polyA tails of different lengths in the sequencing library preparation, which made them ideal for this task. The `ont_polya_standard` dataset comprises six samples with polyA tails of 10, 15, 30, 60, 80, and 100 nucleotides. Similarly, the `eGFP_polyA_DNA`, and `eGFP_polyA_RNA` datasets each contain six samples, with polyA/polyT tails of 10, 30, 40, 60, 100, and 150 nucleotides.

**Evaluation metrics.** We evaluate the performance using two metrics: (1) the polyA tail detection rate, and (2) the mean squared error (MSE) of the polyA length. The output polyA detection flag indicates if a polyA tail is detected for a given read. The detection rate is defined as the proportion of reads with a detected polyA tail. MSE quantifies the difference between the estimated polyA length and that of the ground truth. Higher polyA detection rate and lower MSE mean better performance.

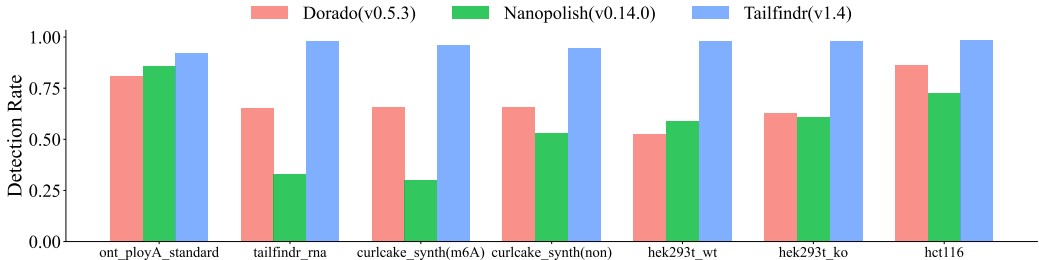

Figure 2: Benchmark of polyA detection rate. Test datasets shown in x-axis.

**Baseline models and benchmark performance.** Three baseline models are selected for comparison: Nanopolish `polya`, Dorado, and Tailfindr. Among these, only Dorado is a deep learning-based method. Figure 2 shows the benchmark results of polyA detection rate on seven test datasets, where Tailfindr demonstrates the best performance across all datasets while Nanopolish `polya` shows large

performance variations across different datasets. Notably, Nanopolish `polya` relies on alignment to the reference genome, whereas Tailfindr and Dorado are alignment-free. All three methods show similar performance in polyA tail length estimation (Appendix Figure 4).

### 4.3 Segmentation and event Alignment (SA) benchmark

The SA task splits the raw signal of a read into segments and aligns the raw signal segments to the reference genome sequence. The aligned $k$mer corresponds to a nucleotide at a specific genomic location that is termed an "event".

**Ground truth acquisition.** Since the SA task is an unsupervised task, there is no ground truth available. However, Oxford Nanopore Technologies has trained their model on a large proprietary dataset and provided a "standard" $k$mer parameter table for some pore versions [31], which contains the mean and standard deviation of raw signal segments aligned to each $k$mer (e.g., 1,024 rows for 5mers). This "standard" $k$mer parameter table can be used in the evaluation.

**Evaluation metrics.** Each SA method generates the alignment of raw signal segments with $k$mers of the reference genome sequence (example results shown in Appendix Figure 5 and Figure 6). For each raw signal segment aligned to a $k$mer $s$ corresponding to a genomic location, we calculate the estimated mean $\mu_s^{(est)}$ and std $\sigma_s^{(est)}$. Two metrics are used to evaluate the performances [25]: (1) the average std $\hat{\sigma}$ over all $k$mers and all reads and (2) the average log-likelihood $\hat{L}$ of the estimated mean $\mu_s^{(est)}$ under the Gaussian distribution with "standard" parameters from ONT, i.e. $\log \mathcal{N}\big(\mu_s^{(ref)}, (\sigma_s^{(ref)})^2\big)$. Detailed formulas are provided in Appendix Methods 4.3. Smaller $\hat{\sigma}$ and larger $\hat{L}$ mean the better performances.

Table 4: Segmentation and event alignment benchmark.

| Test Dataset | Avg. std ($\hat{\sigma}$) ↓ | | | Avg. log p ($\hat{L}$) ↑ | | |
| --- | --- | --- | --- | --- | --- | --- |
| | Nanopolish | Tombo | SegPore | Nanopolish | Tombo | SegPore |
| hek293T_wt | 3.073 | 4.187 | **2.736** | -2.871 | -3.749 | **-2.778** |
| hek293T_ko | 2.948 | 4.204 | **2.670** | -2.856 | -3.749 | **-2.745** |
| hct116 | 3.167 | 4.076 | **2.872** | -2.872 | -3.704 | **-2.746** |

**Baseline models and benchmark performance.** Three baseline models are selected for comparison: Nanopolish `eventalign` (v0.14.0), Tombo `resquiggle` (v1.5.1), and SegPore `eventalign` (v1.0). Table 4 shows that SegPore exhibits the best performance on both average std and log-likelihood.

### 4.4 Modification Detection (MD) benchmark

From the eventalign results of the SA task, we pool all the raw signal segments aligned to the same genomic site. The MD task takes these pooled raw signal segments as input and predicts the modification probability for the given site. Benchmark results are shown for two different RNA modifications, m6A and m5C.

**Ground truth acquisition.** The ground truths are mainly obtained from different short-read sequencing based protocols. Modification rates at different genomic locations are obtained from ChIP-seq protocols where an antibody specifically targeting at a certain RNA modification is used to pull down all reads that possess this RNA modification. For the m6A modification, the ground truth is derived from short-read sequencing data of the following protocols: MeRIP-seq [32], miCLIP [33], and miCLIP2 [34]. For the m5C modification, the ground truth comes from synthetic RNA sequences which use m5C to replace all cytosines (C) in the synthesis, i.e. all C nucleotides in the synthetic RNA sequence have the m5C modification.

**Evaluation metrics.** Given the ground truth and the predicted modification states over all testing genomic sites, we calculate the sensitivity, specificity, precision and recall. Area under the curve (AUC) of the receiver operating characteristic (ROC) curve and precision-recall (PR) curve are used to evaluate the performance of different baseline methods.

**Baseline models and benchmark performance.** For m6A, we choose six baseline methods: Tombo (*de novo*) [35], MINES [36], Nanom6A [37], Epinao [18], SegPore, and m6Anet [38]. For m5C,

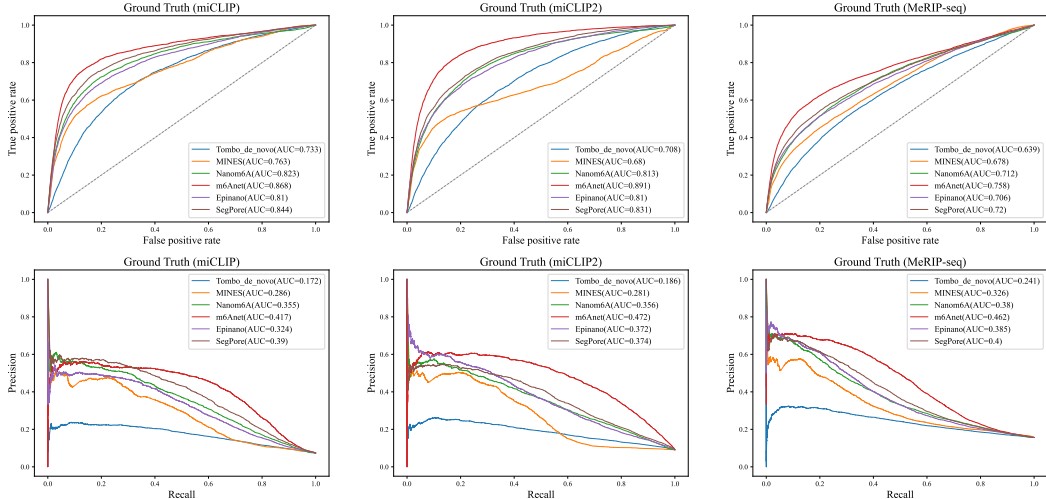

Figure 3: Performances on the m6A modification detection based on three ground truths. The upper panels show the ROC curve, while the lower panels show the PR curve. The test dataset is mESCs_eligos.

we choose three baselines: Tombo (*de novo*), Tombo (alternative model), and CHEUI-solo [39]. Among these, m6Anet and CHEUI are deep learning-based methods. The versions of all models are provided in Appendix Table 4. Figure 3 shows the m6A benchmark results based on three different ground truths. These results suggest that m6Anet has the best performance based on all ground truths. The m5C benchmark results are shown in Appendix Figure 7, where CHEUI-solo has the best performance.

## 5 Software

Beyond the datasets, NanoBaseLib is also a software package that utilizes the pre-processed data by mimicking the standard deep learning neural network training process. The package can pre-process the raw data into the same input and output file formats for different tasks, no matter what tools are used in the preprocessing steps. Based on the preprocessed data, NanoBaseLib uses the abstract classes and functions to load the dataset and calculate the benchmark metrics, significantly reducing the workload for researchers developing new models. NanoBaseLib also provides the full preprocessing pipeline to facilitate data extension. When new Nanopore data becomes available from public repositories, researcher can efficiently run the unified pipeline to incorporate relevant data into NanoBaseLib. The package is available at `https://github.com/nanobaselib/NanoBaseLib`.

## 6 Conclusion

**Limitations.** First, the limitation related to the ground truth arises due to the presence of epigenetic information on RNA molecules, such as RNA modifications. Although these modified bases are represented as the same nucleotide, their current signal levels differ, leading to an inherent upper limit on the accuracy of the ground truth in base calling, where we must rely on approximations. Second, all public datasets in NanoBaseLib are based on R9.4 (or R9.4.1) Nanopore chemistry, and further investigation is required to assess performance on R10 chemistry. We will update NanoBaseLib as new datasets become available. Therefore, the current version, including the benckmarks and datasets, is designated as *NanoBaseLib v1.0*. Third, the benchmarking experiments conducted in this work used only a subset of the 16 available datasets to demonstrate NanoBaseLib's capabilities. However, researchers can achieve more comprehensive results by utilizing the remaining datasets in *NanoBaseLib v1.0*. Lastly, potential negative societal impacts are discussed in Appendix Methods 4.4.

**Summary.** The core concept of NanoBaseLib is "One Dataset, Multiple Tasks", which enables the analysis of various tasks using the same dataset. This approach contrasts with traditional methods, where multiple datasets are processed separately for individual tasks. From a dataset perspective, NanoBaseLib integrates 16 public datasets comprising over 30 million reads, with a unified preprocessing pipeline and standardized storage formats that contribute to fair benchmarking. From a task perspective, NanoBaseLib allows multiple tasks to be performed on the same datasets, enhancing the efficiency of dataset reuse. Additionally, we provide a software package to facilitate easier and more accurate analysis of newly available datasets. We hope that NanoBaseLib significantly reduces the challenges of bioinformatic preprocessing and attracts more machine learning researchers to the Nanopore sequencing field.

## Acknowledgments and Disclosure of Funding

We acknowledge the computational resources provided by the Aalto Science-IT project and CSC – IT Center for Science, Finland. We also acknowledge Aki Vehtari, Bo Zhao, Ville Hyvönen, Yue Jiang from Aalto University, as well as Jie Tan from The Chinese University of Hong Kong, for their valuable comments on the manuscript. We thank Research Council of Finland grants (NO. 335858, 358086) to GC, CF and LC.

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
