# NanoBaseLib: A Multi-Task Benchmark Dataset for Nanopore Sequencing *Supplementary Material*

**Guangzhao Cheng**[1] **Chengbo Fu**[1] **Lu Cheng**[1,2,*]

[1] Department of Computer Science, Aalto University, Finland
[2] Institute of Biomedicine, University of Eastern Finland, Finland

# Contents

*Correspondence to: lu.cheng.ac@gmail.com.

# 1 Dataset Checklist

1. Submission introducing new datasets must include the following in the supplementary materials:

   (a) Dataset documentation and intended uses. Recommended documentation frameworks include datasheets for datasets, dataset nutrition labels, data statements for NLP, and accountability frameworks. **A: Available on the dataset website (https://nanobaselib.github.io/dataset.html).**

   (b) URL to website/platform where the dataset/benchmark can be viewed and downloaded by the reviewers. **A: The raw data download instruction: https://nanobaselib.github.io/raw.html. The benchmarks and processed datasets are available at https://doi.org/10.5281/zenodo.10889896.**

   (c) URL to Croissant metadata record documenting the dataset/benchmark available for viewing and downloading by the reviewers. You can create your Croissant metadata using e.g. the Python library available here: https://github.com/mlcommons/croissant **A: NA.**

   (d) Author statement that they bear all responsibility in case of violation of rights, etc., and confirmation of the data license. **A: Yes.**

   (e) Hosting, licensing, and maintenance plan. The choice of hosting platform is yours, as long as you ensure access to the data (possibly through a curated interface) and will provide the necessary maintenance. **A: The dataset is hosted on Zenodo. The website is hosted on GitHub, where it will be maintained and regularly updated.**

2. To ensure accessibility, the supplementary materials for datasets must include the following:

   (a) Links to access the dataset and its metadata. This can be hidden upon submission if the dataset is not yet publicly available but must be added in the camera-ready version. In select cases, e.g when the data can only be released at a later date, this can be added afterward. Simulation environments should link to (open source) code repositories. **A: https://nanobaselib.github.io or https://doi.org/10.5281/zenodo.10889896.**

   (b) The dataset itself should ideally use an open and widely used data format. Provide a detailed explanation on how the dataset can be read. For simulation environments, use existing frameworks or explain how they can be used. **A: Available on the dataset website.**

   (c) Long-term preservation: It must be clear that the dataset will be available for a long time, either by uploading to a data repository or by explaining how the authors themselves will ensure this. **A: Zenodo is a long-time storage open repository.**

   (d) Explicit license: Authors must choose a license, ideally a CC license for datasets, or an open source license for code (e.g. RL environments). **A: The processed dataset is licensed under CC BY license.**

   (e) Add structured metadata to a dataset's meta-data page using Web standards (like schema.org and DCAT): This allows it to be discovered and organized by anyone. If you use an existing data repository, this is often done automatically. **A: The dataset structure is available on the dataset website (https://nanobaselib.github.io/dataset.html).**

   (f) Highly recommended: a persistent dereferenceable identifier (e.g. a DOI minted by a data repository or a prefix on identifiers.org) for datasets, or a code repository (e.g. GitHub, GitLab,...) for code. If this is not possible or useful, please explain why. **A: DOI: 10.5281/zenodo.10889896.**

3. For benchmarks, the supplementary materials must ensure that all results are easily reproducible. Where possible, use a reproducibility framework such as the ML reproducibility checklist, or otherwise guarantee that all results can be easily reproduced, i.e. all necessary datasets, code, and evaluation procedures must be accessible and documented. **A: The code for the benchmarks is available at GitHub (https://github.com/nanobaselib/NanoBaseLib).**

4. For papers introducing best practices in creating or curating datasets and benchmarks, the above supplementary materials are not required. **A: NA.**

# 2 Appendix Figures

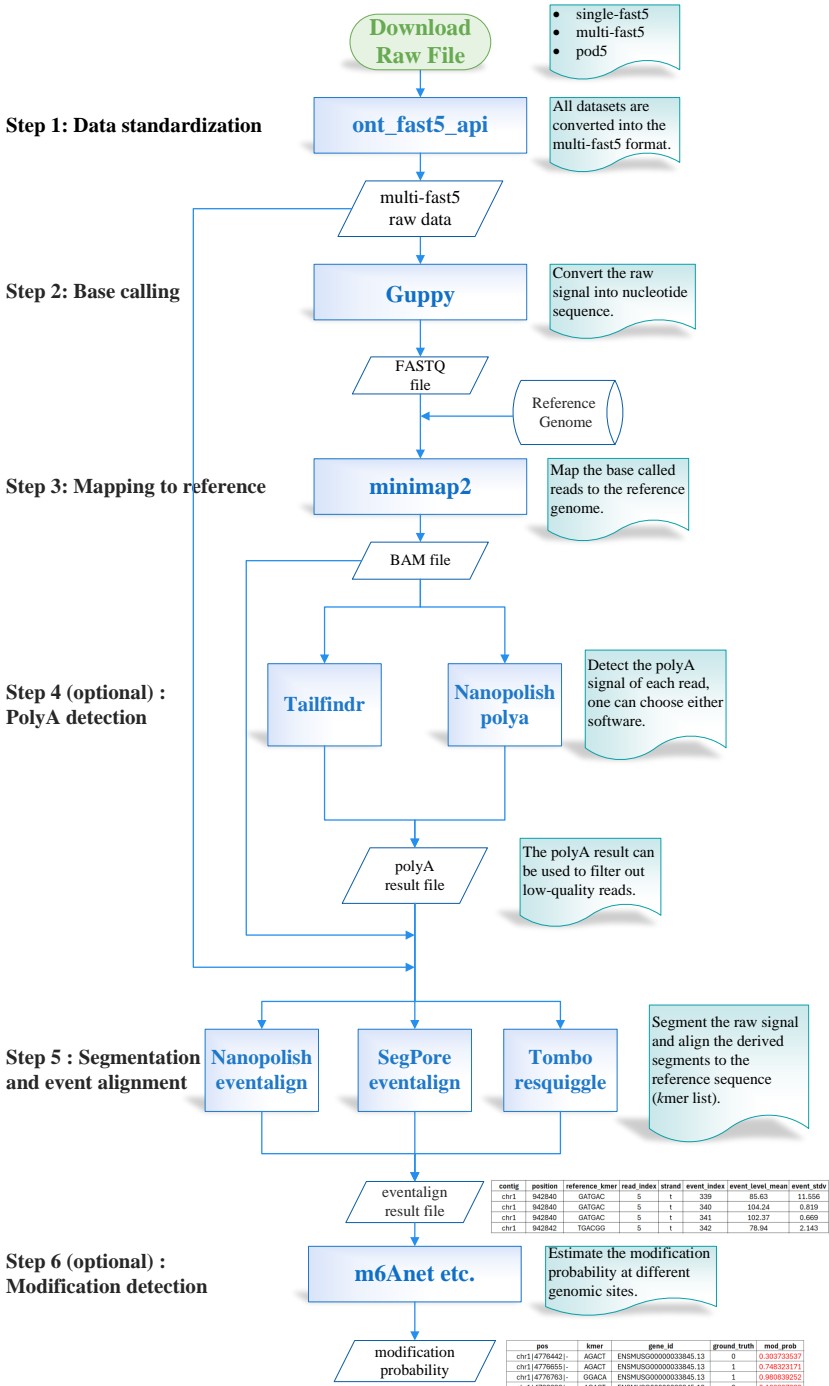

Figure 1: NanoBaseLib dataset processing workflow.

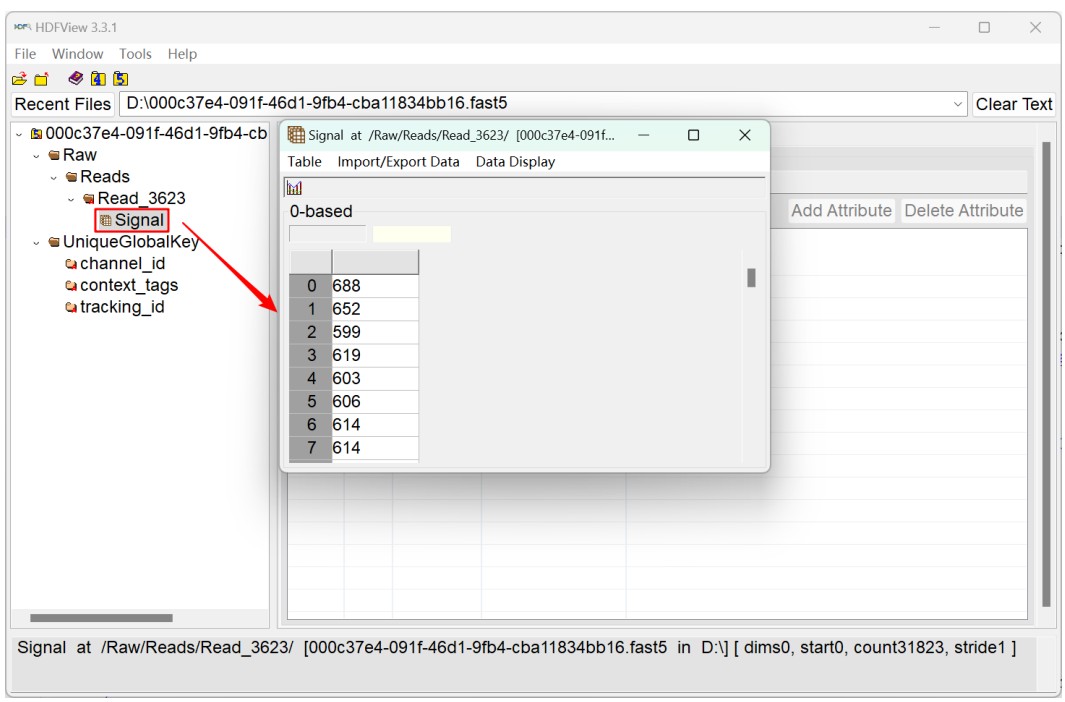

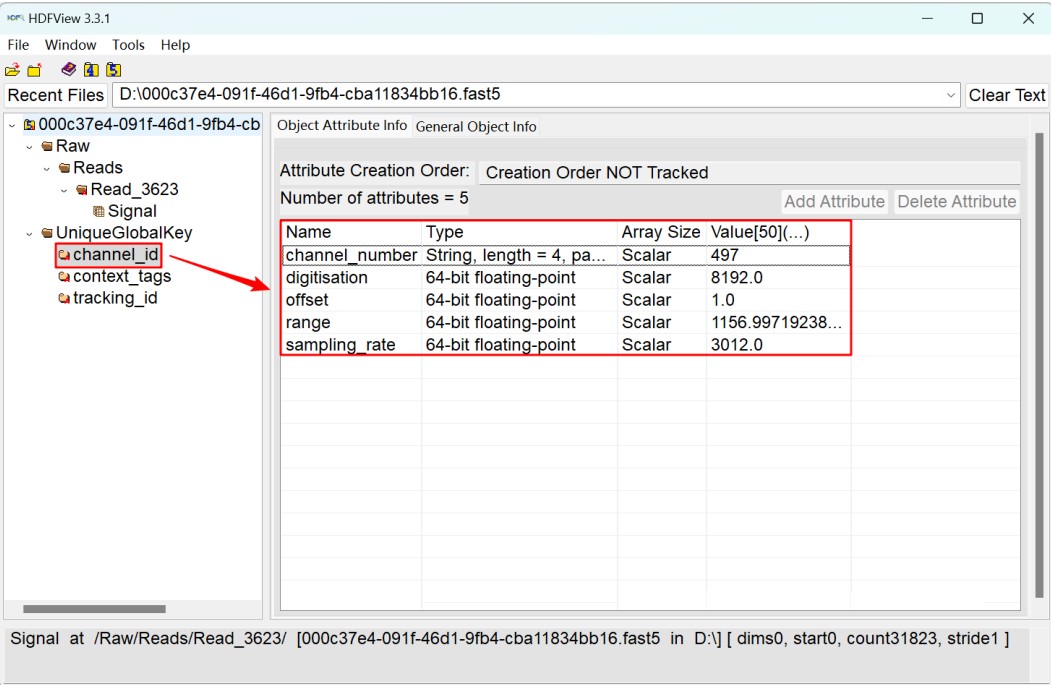

Figure 2: Illustration of the single fast5 file. The top panel displays the raw signal, while the bottom panel presents some meta information.

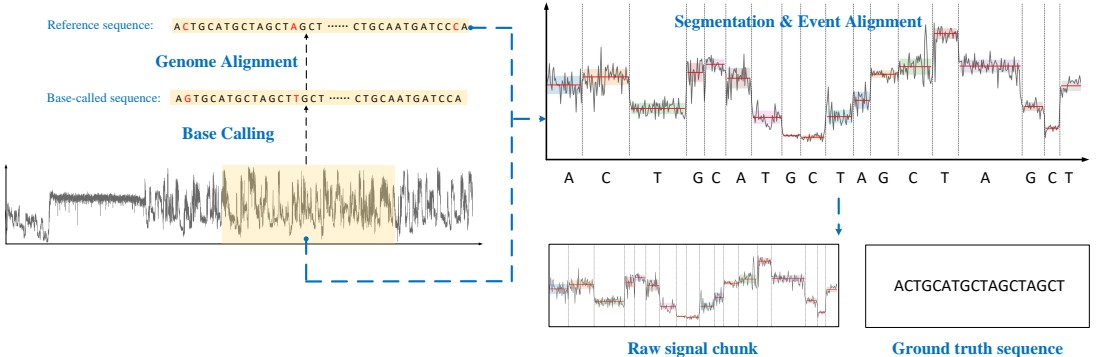

Figure 3: Illustration of Base Calling (BC) task ground truth acquisition. We perform base calling (using Guppy v6.0.1) firstly, then align the base-called sequence with the reference genome and run Nanopolish "eventalign". Finally, we extract the matched raw signal segments and reference sequence fragments as the ground truth.

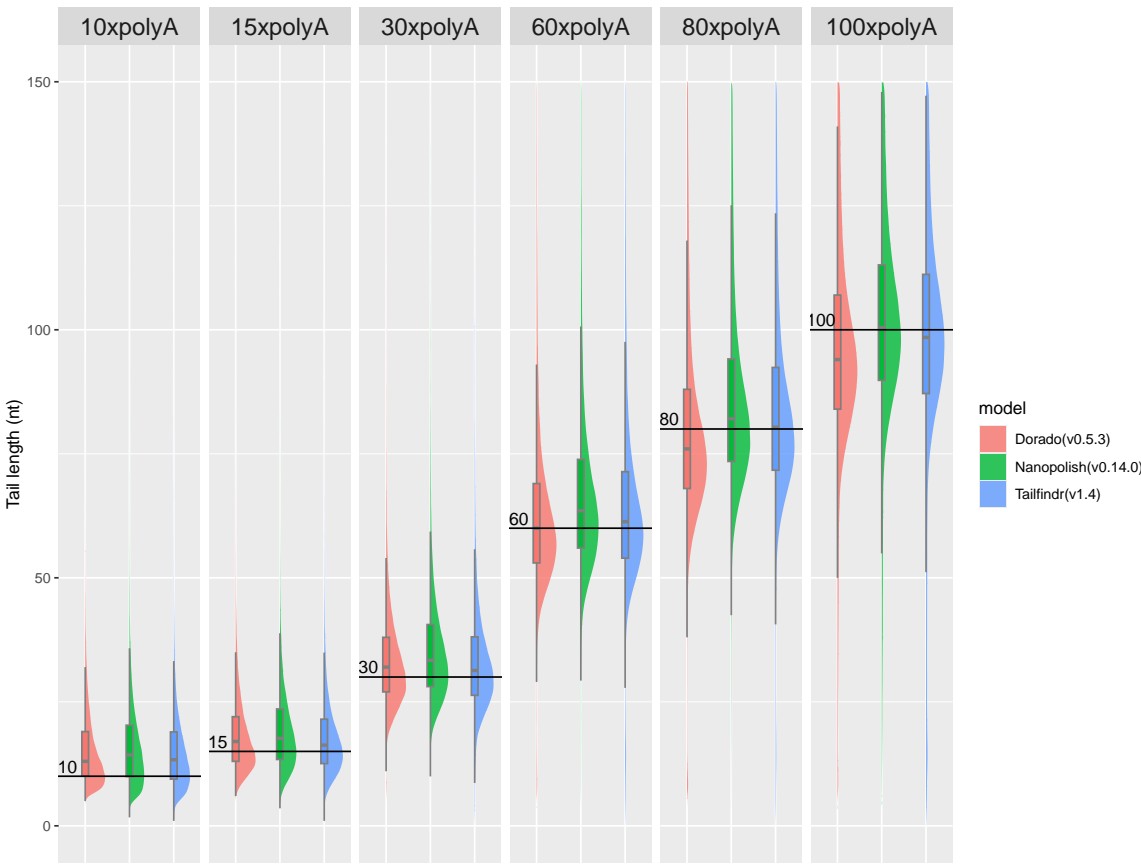

Figure 4: PolyA tail length distribution on test dataset ont_polya_standard. The black line and the numbers (10, 15, 30, 60, 80, 100) represent the ground truth. The results are based on Nanopolish (v0.14), Tailfindr (v1.4), and Dorado (v0.5.3).

| contig | position | reference_kmer | read_index | strand | event_index | event_level_mean | event_stdv | event_length | model_kmer | model_mean | model_stdv | standardized_level | start_idx | end_idx |
|---|---|---|---|---|---|---|---|---|---|---|---|---|---|---|
| U00096.3| | 0 | ATGTCC | 4 | t | 81 | 71.02 | 0.582 | 0.00075 | ATGTCC | 81.5 | 2.83 | -2.81 | 429 | 432 |
| U00096.3| | 0 | ATGTCC | 4 | t | 82 | 73.3 | 0.793 | 0.00075 | ATGTCC | 81.5 | 2.83 | -2.20 | 432 | 435 |
| U00096.3| | 0 | ATGTCC | 4 | t | 83 | 72.07 | 0.22 | 0.00075 | ATGTCC | 81.5 | 2.83 | -2.53 | 435 | 438 |
| U00096.3| | 0 | ATGTCC | 4 | t | 84 | 73.07 | 0.855 | 0.001 | ATGTCC | 81.5 | 2.83 | -2.26 | 438 | 442 |
| U00096.3| | 0 | ATGTCC | 4 | t | 85 | 70.4 | 0.634 | 0.00125 | NNNNNN | 0 | 0 | inf | 442 | 447 |
| U00096.3| | 3 | TCCGTA | 4 | t | 86 | 94.26 | 4.994 | 0.00175 | TCCGTA | 91.55 | 2.13 | 0.97 | 447 | 454 |
| U00096.3| | 4 | CCGTAG | 4 | t | 87 | 76.73 | 1.508 | 0.0015 | CCGTAG | 81.09 | 2.14 | -1.55 | 454 | 460 |
| U00096.3| | 4 | CCGTAG | 4 | t | 88 | 79.92 | 0.876 | 0.00075 | CCGTAG | 81.09 | 2.14 | -0.42 | 460 | 463 |
| U00096.3| | 4 | CCGTAG | 4 | t | 89 | 77.17 | 0.465 | 0.00075 | CCGTAG | 81.09 | 2.14 | -1.39 | 463 | 466 |
| U00096.3| | 4 | CCGTAG | 4 | t | 90 | 78.36 | 2.252 | 0.00225 | CCGTAG | 81.09 | 2.14 | -0.97 | 466 | 475 |
| U00096.3| | 5 | CGTAGA | 4 | t | 91 | 99.58 | 2.077 | 0.00075 | CGTAGA | 104.43 | 2.78 | -1.33 | 475 | 478 |
| U00096.3| | 5 | CGTAGA | 4 | t | 92 | 100.22 | 2.018 | 0.002 | CGTAGA | 104.43 | 2.78 | -1.15 | 478 | 486 |
| U00096.3| | 6 | GTAGAA | 4 | t | 93 | 84.06 | 0.786 | 0.00125 | GTAGAA | 89.37 | 2.32 | -1.74 | 486 | 491 |
| U00096.3| | 7 | TAGAAA | 4 | t | 94 | 81.43 | 1.959 | 0.00125 | TAGAAA | 80.56 | 1.83 | 0.36 | 491 | 496 |

Figure 5: Illustration of Nanopolish "eventalign" output

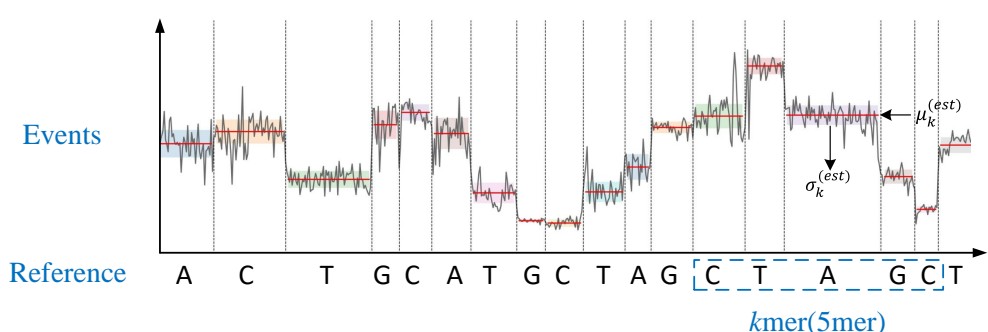

Figure 6: Illustration of segmentation and event alignment.

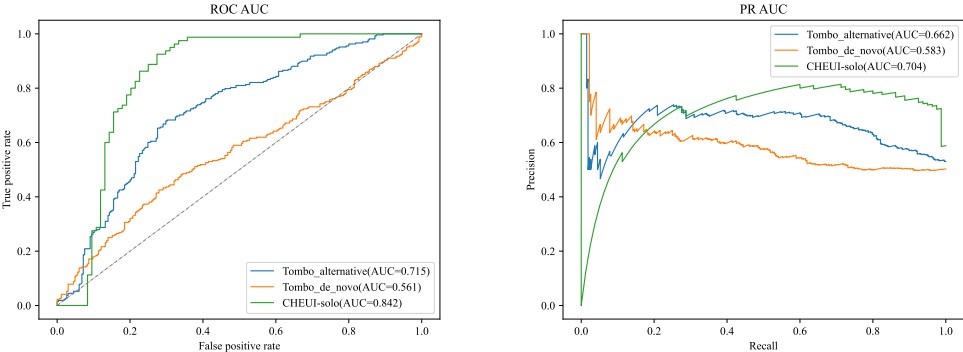

Figure 7: m5C modification detection benchmark on test dataset ecoli_eligos (IVT_m5C and IVT_normalC).

# 3 Appendix Tables

Table 1: NanoBaseLib dataset comprehensive information.

| Dataset | Accession | Sample | Kit | Flowcell |
|---|---|---|---|---|
| ont_polya_standard | PRJEB28423 | 10xpolyA | rna001 | flo-min106 |
| | | 15xpolyA | rna001 | flo-min106 |
| | | 30xpolyA | rna001 | flo-min106 |
| | | 60xpolyA | rna001 | flo-min106 |
| | | 80xpolyA | rna001 | flo-min106 |
| | | 100xpolyA | rna001 | flo-min106 |
| eGFP_polyA_DNA | PRJEB31806 | dna_rep1_sqklsk108_flipflop | lsk108 | flo-min106 |
| | | dna_rep2_sqklsk109_flipflop | lsk109 | flo-min106 |
| eGFP_polyA_RNA | PRJEB31806 | rna_rep1_sqkrna001_plus_rt | rna001 | flo-min106 |
| | | rna_rep2_sqkrna001_plus_rt | rna001 | flo-min106 |
| | | rna_rep3_sqkrna002_minus_rt | rna002 | flo-min106 |
| lambda_phage | PRJNA926802 | VER5940 | lsk109 | flo-flg001 |
| NA12878 | PRJEB23027 | FAB42828 | lsk108 | flo-min106 |
| | | FAF04090 | lsk108 | flo-min106 |
| | | FAF09968 | lsk108 | flo-min106 |
| curlcake | PRJNA511582 | m6A-mod-rep1 | rna001 | flo-min106 |
| | | m6A-mod-rep2 | rna001 | flo-min106 |
| | | non-mod-rep1 | rna001 | flo-min106 |
| | | non-mod-rep2 | rna001 | flo-min106 |
| scBY4741_m5C | PRJNA563591 | m5C_modified | rna001 | flo-min106 |
| scBY4741_hm5C | PRJNA548268 | hm5C_modified | rna001 | flo-min106 |
| scBY4741_pU | PRJNA549001 | pU_modified | rna001 | flo-min106 |
| hct116 | PRJEB44348 | HCT-WT-rep1 | rna002 | flo-min106 |
| | | HCT-WT-rep2 | rna002 | flo-min106 |
| | | HCT-WT-rep3 | rna002 | flo-min106 |
| hek293t_wt | PRJEB40872 | HEK293T-WT-rep1 | rna001 | flo-min106 |
| | | HEK293T-WT-rep2 | rna002 | flo-min106 |
| | | HEK293T-WT-rep3 | rna002 | flo-min106 |
| hek293t_ko | PRJEB40872 | HEK293T-WT-rep1 | rna001 | flo-min106 |
| | | HEK293T-WT-rep2 | rna002 | flo-min106 |
| | | HEK293T-WT-rep3 | rna002 | flo-min106 |
| mESCs_eligos | PRJNA497103 | mESCs_Mettl3_WT | rna002 | flo-min106 |
| | | mESCs_Mettl3_KO | rna002 | flo-min106 |
| ecoli_eligos | PRJNA497103 | IVT_Inosine | rna002 | flo-min106 |
| | | IVT_m5C | rna002 | flo-min106 |
| | | IVT_m6A | rna002 | flo-min106 |
| | | IVT_normalA | rna002 | flo-min106 |
| | | IVT_normalC | rna002 | flo-min106 |
| dinopore_ivt | SRP363295 | gBlock_pureI | rna001 | flo-min106 |
| | | gBlock_G | rna001 | flo-min106 |
| dinopore_xenopus | SRP363295 | rep3_stage1_20200812 | rna002 | flo-min106 |
| | | rep3_stage1_20201005 | rna002 | flo-min106 |
| | | rep3_stage9_20200812 | rna002 | flo-min106 |
| | | rep3_stage9_20201008 | rna002 | flo-min106 |

Table 2: NanoBaseLib dataset statistics. Avg. $L_{signal}$ represents the average raw current signal length. Avg. $L_{base}$ represents the average base sequence length, which are from Guppy 6.0.1.

| Dataset | Type | Sample | #Reads | Avg. $L_{signal}$ | Avg. $L_{base}$ |
|---|---|---|---|---|---|
| ont_polya_standard | RNA | 10xpolyA | 92,428 | 59001.85 | 1207.22 |
| | | 15xpolyA | 91,084 | 56518.49 | 1216.28 |
| | | 30xpolyA | 63,886 | 54111.54 | 1192.65 |
| | | 60xpolyA | 108,314 | 57397.07 | 1172.57 |
| | | 80xpolyA | 409,634 | 47166.28 | 859.32 |
| | | 100xpolyA | 279,895 | 61938.01 | 1173.39 |
| eGFP_polyA_DNA | cDNA | dna_rep1_sqklsk108_flipflop | 484,000 | 8956.69 | 763.46 |
| | | dna_rep2_sqklsk109_flipflop | 280,428 | 21619.23 | 1667.14 |
| eGFP_polyA_RNA | RNA | rna_rep1_sqkrna001_plus_rt | 922,826 | 57068.67 | 1126.53 |
| | | rna_rep2_sqkrna001_plus_rt | 1,452,042 | 50103.37 | 928.41 |
| | | rna_rep3_sqkrna002_minus_rt | 592,571 | 30888.61 | 465.02 |
| lambda_phage | DNA | VER5940 | 113,514 | 116272.62 | 9561.99 |
| NA12878 | DNA | FAB42828 | 33,633 | 131148.91 | 6810.35 |
| | | FAF04090 | 62,833 | 509826.89 | 17801.22 |
| | | FAF09968 | 21,947 | 334920.97 | 53615.01 |
| curlcake | RNA | m6A-mod-rep1 | 134,374 | 69745.77 | 850.16 |
| | | m6A-mod-rep2 | 638,860 | 58341.88 | 835.01 |
| | | non-mod-rep1 | 66,736 | 57930.60 | 866.98 |
| | | non-mod-rep2 | 846,595 | 61719.51 | 1066.53 |
| scBY4741_m5C | RNA | m5C_modified | 415,453 | 40792.42 | 539.89 |
| scBY4741_hm5C | RNA | hm5C_modified | 111,015 | 81528.20 | 1022.88 |
| scBY4741_pU | RNA | pU_modified | 42,386 | 46652.89 | 475.18 |
| hct116 | RNA | HCT-WT-rep1 | 987,488 | 66363.12 | 1217.43 |
| | | HCT-WT-rep2 | 1,015,893 | 57524.51 | 1023.03 |
| | | HCT-WT-rep3 | 1,673,394 | 65628.29 | 1153.23 |
| hek293t_wt | RNA | HEK293T-WT-rep1 | 1,040,661 | 60169.77 | 939.80 |
| | | HEK293T-WT-rep2 | 1,396,000 | 54077.71 | 1077.61 |
| | | HEK293T-WT-rep3 | 513,561 | 56785.55 | 1005.06 |
| hek293t_ko | RNA | HEK293T-WT-rep1 | 1,490,210 | 58140.70 | 952.63 |
| | | HEK293T-WT-rep2 | 1,815,589 | 52569.78 | 993.85 |
| | | HEK293T-WT-rep3 | 1,677,075 | 50185.96 | 970.32 |
| mESCs_eligos | RNA | mESCs_Mettl3_WT | 3,163,286 | 33202.35 | 526.23 |
| | | mESCs_Mettl3_KO | 1,527,561 | 28350.74 | 437.70 |
| ecoli_eligos | RNA | IVT_Inosine | 811,953 | 32978.04 | 845.43 |
| | | IVT_m5C | 573,674 | 45397.06 | 719.52 |
| | | IVT_m6A | 1,482,437 | 41642.13 | 708.29 |
| | | IVT_normalA | 383,209 | 33499.75 | 620.83 |
| | | IVT_normalC | 452,806 | 44566.76 | 731.75 |
| dinopore_ivt | RNA | gBlock_pureI | 165,628 | 29869.74 | 450.32 |
| | | gBlock_G | 150,405 | 32047.08 | 641.17 |
| dinopore_xenopus | RNA | rep3_stage1_20200812 | 1,451,289 | 46688.45 | 917.23 |
| | | rep3_stage1_20201005 | 1,812,200 | 27213.72 | 532.63 |
| | | rep3_stage9_20200812 | 1,560,032 | 44621.79 | 894.37 |
| | | rep3_stage9_20201008 | 1,251,130 | 31185.45 | 448.15 |

Table 3: NanoBaseLib task overview.

| Task | Input | Output | Category | Typical Model |
|------|-------|--------|----------|---------------|
| Base calling (BC) | Raw current signal sequence | Nucleotide sequence | Supervised Learning, Generative Model | CNN + LSTM + CTC-CRF [1] 
 UNet + GRU + CE [2] 
 CNN + Transformer + CTC [3] 
 ResNet CNN + CTC [4] |
| PolyA detection (PD) | Raw current signal sequence | PolyA tail length and borders | Unsupervised or Supervised Learning, Predictive Model | Hidden Markov model[5, 6] |
| Segmentation and event alignment (SA) | Raw signal and reference sequence | Event alignment results | Unsupervised Learning, Predictive Model | Hidden Markov model [7, 8] |
| Modification detection (MD) | Event alignment results | Modification probability for each site & read | Supervised Learning, Predictive Model, Multiple Instance Learning | SVM [9] 
 CNN [10] |

Table 4: Data preprocessing software overview and limitations.

| Software | Version | Link | Limitation |
|----------|---------|------|------------|
| bedtools | 2.30.3 | https://bedtools.readthedocs.io/en/latest/ | NA |
| Bonito | 0.7.3 | https://github.com/nanoporetech/bonito | NA |
| causalcall | NA | https://github.com/scutbioinformatic/causalcall | Only for DNA |
| CHEUI | NA | https://github.com/comprna/CHEUI | m6A and m5C |
| Dorado | 0.5.3 | https://community.nanoporetech.com/downloads | NA |
| Epinano | 1.2.0 | https://github.com/novoalab/EpiNano | NA |
| Guppy | 6.0.1 | https://community.nanoporetech.com/downloads | NA |
| h5py | 1.8.18 | https://www.h5py.org/ | NA |
| minimap2 | 2.24 | https://github.com/lh3/minimap2 | NA |
| MINES | NA | https://github.com/YeoLab/MINES | Only for m6A |
| m6Anet | 1.0 | https://github.com/GoekeLab/m6anet | Only for m6A |
| Nanopolish | 0.14.0 | https://github.com/jts/nanopolish | NA |
| Nanom6A | 2.0 | https://github.com/gaoyubang/nanom6A | Only for m6A |
| ont-fast5-api | 4.0.2 | https://pod5-file-format.readthedocs.io | NA |
| Rodan | NA | https://github.com/biodlab/RODAN | Only for RNA |
| SegPore | 1.0 | https://github.com/guangzhaocs/SegPore | Only for RNA |
| Tailfindr | 1.4 | https://github.com/adnaniazi/tailfindr | NA |
| Tombo | 1.5.1 | https://nanoporetech.github.io/tombo | NA |

Table 5: ONT basecaller configuration parameters

| Basecaller | Version | Sample | Configure |
|------------|---------|--------|-----------|
| Guppy | 2.3.1 | DNA | dna_r9.4.1_450bps.cfg |
| Guppy | 2.3.1 | RNA | rna_r9.4.1_70bps.cfg |
| Guppy | 4.5.4 | DNA | dna_r9.4.1_450bps_hac.cfg |
| Guppy | 4.5.4 | RNA | rna_r9.4.1_70bps_hac.cfg |
| Guppy | 6.0.1 | DNA | dna_r9.4.1_450bps_hac.cfg |
| Guppy | 6.0.1 | RNA | rna_r9.4.1_70bps_hac.cfg |
| Bonito | 0.7.3 | DNA | dna_r9.4.1_e8_hac@v3.3 |
| Dorado | 0.5.3 | DNA | dna_r9.4.1_e8_hac@v3.3 |
| Dorado | 0.7.0 | DNA | dna_r9.4.1_e8_hac@v3.3 |
| Dorado | 0.5.3 | RNA | rna002_70bps_hac@v3 |
| Dorado | 0.7.0 | RNA | rna002_70bps_hac@v3 |

# 4 Appendix Methods

## 4.1 Pros and Cons of Nanopore Sequencing

**Pros.** Compared with Illumina sequencing, Nanopore sequencing has three advantages. First, it sequences long reads (10-100 kb), where a "read" refers to a measured DNA/RNA sequence composed of the nucleotide bases adenine (A), cytosine (C), guanine (G), thymine (T) or uracil (U). The read length of Illumina sequencing is less than 300 bp, which is ∼1% of a long read. Long reads greatly facilitate genome assembly by reducing ambiguities in the genome backbone, especially in repetitive genomic regions. Second, the sequencing library preparation is generally simpler than second-generation sequencing, which makes it ideal for pathogen surveillance in wild environments. Third, the current signals directly carry chemical information on DNA and RNA molecules, e.g., DNA/RNA modifications, which makes it perfect for epi-genetic applications, such as epi-genetic disease studies [11], RNA structures [12], and mRNA vaccine development [13].

**Cons.** The most significant disadvantage of Nanopore sequencing compared with Illumina sequencing is its low accuracy in base calling. The average base calling accuracy of Nanopore RNA sequencing is around 90% [14], while the accuracy is 99.9% for Illumina cDNA sequencing [15]. This makes Nanopore sequencing not ideal for many classic biological applications, e.g., detection of single nucleotide variants (SNVs). The costs of Nanopore sequencing are higher than Illumina sequencing, while the throughput is lower. Also, current methods could not provide a satisfactory alignment of the raw current signal and the reference sequences, which limits its performance in DNA/RNA modification estimation. In short, there is plenty of room for improvement in various computational tasks of Nanopore sequencing data analysis. We believe these disadvantages could be overcome by developing new machine learning models.

## 4.2 Raw signal processing and normalization

Appendix Figure 2 illustrates the structure of a single-fast5 file from Nanopore sequencing. In the above panel, the "Signal" data represents the raw current passing through the pore (type: int16). Oxford Nanopore Technology employs different pores (proteins) in various products. A single flow cell in the sequencing device can contain between 512 and 2675 pores, each referred to as a channel. As shown in the below panel, the fast5 file also includes attributes associated with the channel through which the read passes. These parameters include `channel_number` (the channel number from which the read was acquired), `digitisation` (the digitisation is the number of quantisation levels in the Analog to Digital Converter (ADC)), `offset` (the ADC offset error), `range` (the full scale measurement range in pico amperes), and `sampling_rate` (sampling frequency of the ADC).

The raw signal can be converted into pico Ampere (pA) values using attributes available in the `channel_id` group by the equation:

$$signal\_in\_pico\_ampere = \frac{(raw\_signal\_value + offset) \times range}{digitisation}. \tag{1}$$

To improve the accuracy of analyses for various downstream tasks, some tools aim to further standardize or normalize the signal (pA).

**Tombo.** Tombo uses median shift and MAD (median absolute deviation) scale parameters to normalize the signal [16]:

$$norm\_signal = \frac{signal\_in\_pico\_ampere - median}{MAD}. \tag{2}$$

**Nanopolish.** For each read, Nanopolish estimates a scale parameter to standardize the signal. You can add the `--scale-events` option in the nanopolish `eventalign` command to enable this feature [5].

**SegPore.** SegPore first detects the polyA tail and calculates its mean ($\mu_{polyA}$) and standard deviation ($\sigma_{polyA}$). These values are then used to standardize the signal [7]:

$$stand\_signal = \frac{signal\_in\_pico\_ampere - \mu_{polyA}}{\sigma_{polyA}} \times \sigma_{stand\_polyA} + \mu_{stand\_polyA} \tag{3}$$

where the $\mu_{stand\_polyA}$ and $\sigma_{stand\_polyA}$ represent the mean and standard deviation of the kmer "AAAAA" from ONT's standard kmer table [17].

## 4.3 Segmentation and event alignment

Appendix Figure 5 is an example of the Nanopolish `eventalign` output, which illustrates the segmentation and event alignment results. Each line represents an "event", which contains mapped chromosome/transcript $t$ (`contig`), the location on the chromosome/transcript $p$ (`position`), the corresponding kmer $s$ (`reference_kmer`), the estimated mean of this "event" $\mu_s^{(est)}$ (`event_level_mean`), the estimated std of this "event" $\sigma_s^{(est)}$ (`event_stdv`), `model_mean`, `model_stdv`, and other information. Columns `model_mean` and `model_stdv` are from a standard kmer parameter table, which is provided by ONT [17]. This kmer table contains the standard mean $\mu_s^{(ref)}$ and std $\sigma_s^{(ref)}$ for each kmer $s$. For example, the standard mean is 81.5 and std is 2.83 for kmer `ATGTCC` (Row $1 \sim 4$ in Appendix Figure 5).

Given the eventalign results, we use two metrics to evaluate the performance of the segmentation and event alignment task: (1) the average std $\hat{\sigma}$, and (2) the average log-likelihood $\hat{L}$. Assuming there are $N$ reads in total, and the $n$th read has $K_n$ events. We denote the $k$th event of the $n$th read by $e_{n,k} = \left(t_{n,k}, p_{n,k}, s_{n,k}, \mu_{s_{n,k}}^{(est)}, \sigma_{s_{n,k}}^{(est)}, \mu_{s_{n,k}}^{(ref)}, \sigma_{s_{n,k}}^{(ref)}, \cdots\right)$, where $t_{n,k}$ is the mapped chromosome/transcript, $p_{n,k}$ is the mapped genomic location, $s_{n,k}$ is the corresponding kmer. The average std $\hat{\sigma}$ is defined as

$$\hat{\sigma} = \frac{1}{N} \sum_{n=1}^{N} \left\{ \frac{1}{K_n} \sum_{k=1}^{K_n} \sigma_{s_{n,k}}^{(est)} \right\} \tag{4}$$

and the average log-likelihood $\hat{L}$ is defined as

$$\hat{L} = \frac{1}{N} \sum_{n=1}^{N} \left\{ \frac{1}{K_n} \sum_{k=1}^{K_n} \log \mathcal{N}\left(\mu_{s_{n,k}}^{(est)} | \mu_{s_{n,k}}^{(ref)}, \sigma_{s_{n,k}}^{(ref)}\right) \right\} \tag{5}$$

As shown in Appendix Figure 6, the red line represents the event mean $\mu_k^{(est)}$ and the shaded area represents the std $\sigma_k^{(est)}$ for event $k$. A poorly segmented raw signal corresponding to an event will exhibit a large standard deviation. Therefore, a smaller $\hat{\sigma}$ indicates lower variations within the raw signal segment, signifying better performance.

If an event is aligned to the correct reference kmer, the mean $\mu_{n,k}^{(est)}$ will be close to the reference $\mu_{s_{n,k}}^{(ref)}$ and the log-likelihood $\hat{L}$ will be large. So higher $\hat{L}$ means more similar results to ONT's estimates and better performances.

## 4.4 Potential negative societal impact

All raw datasets were collected from public resources, and we only preprocessed them using a standard pipeline. However, the datasets could still be exploited for unforeseen purposes, such as commercialization. In which, the anonymized genetic data might be used by companies to develop targeted products, potentially compromising privacy or leading to social inequities, like expensive drugs limited to specific populations. Additionally, the data might be repurposed for unintended research, possibly contradicting the original data providers' intent or causing societal harm.