# OpenReview forum: "NanoBaseLib: A Multi-Task Benchmark Dataset for Nanopore Sequencing"
_NeurIPS.cc/2024/Datasets_and_Benchmarks_Track — NeurIPS 2024 Track Datasets and Benchmarks Poster_

### Official Review · Reviewer_bhmA · 2024-07-21
**An awesome benchmark for applying machine learning to nanopore sequencing**

**Rating:** 7
**Confidence:** 3
**Correctness:** The claims are correct. The dataset i…
**Clarity:** Yes.

**Review:**

Quality:
This work presents a high-quality benchmark dataset, with good background and task description and detailed documentation. The quality can be further improved by benchmarking deep learning based baseline methods.

Clarity:
The writing is clear and well-organized.

Originality:
This work makes an excellent novelty contribution of proposing a novel dataset.

Significance:
This work makes significant and sound contributions in facilitate the application of machine learning to nanopore sequencing.

**Strengths:**

- Propose a novel dataset for nanopore sequencing.
- Good paper writing and organization.
- Detailed and informative background introduction and dataset documentation.

**Additional Feedback:**

No additional feedback.

**Documentation:**

There is sufficient detail on data collection and organization, availability and maintenance, and ethical and responsible use.

**Ethics:**

No ethical concerns.

**Limitations:**

The authors have adequately addressed the limitations and potential negative societal impact of their work.

**Opportunities For Improvement:**

Authors are recommended to benchmark the performance of at least one deep learning baseline method for each dataset in NanoBaseLib.

**Relation To Prior Work:**

Yes.

**Summary And Contributions:**

This paper proposes NanoBaseLib, a novel benchmark dataset for applying machine learning to nanopore sequencing. NanoBaseLib integrates 16 datasets for four major tasks in nanopore sequencing.

---

> ### Author Rebuttal · Authors · 2024-08-16
>
> Dear Reviewer #bhmA:
>
> Thank you for your constructive comments! We would like to address your suggestions regarding the deep learning baseline methods here.
>
> > Authors are recommended to benchmark the performance of at least one deep learning baseline method for each dataset in NanoBaseLib.
>
> NanoBaseLib is designed to incorporate four key tasks: Base Calling (BC), PolyA Detection (PD), Segmentation and Event Alignment (SA), and Modification Detection (MD). While some tasks, such as BC and MD, are typically addressed using deep learning methods, others, like PD and SA, are more commonly approached with traditional machine learning methods.
>
> In NanoBaseLib, the Base Calling (BC), Modification Detection (MD), and PolyA Detection (PD) tasks all include at least one deep learning-based method. Specifically, all baselines for Base Calling are deep learning methods. In the Modification Detection task, both m6Anet and CHEUI are based on deep learning. For PolyA Detection, Dorado is a deep learning method. We will clarify these details in the corresponding subsections of Section 4.
>
> The Segmentation and Event Alignment (SA) task is essentially an unsupervised task, with the primary challenge being the lack of high-quality labels. Therefore the commonly adopted methods (Tombo, Nanopolish, SegPore) for this task are generally based on HMMs. To date, we are not aware of any deep learning-based approaches for this task. However, we would be eager to incorporate such a method into NanoBaseLib should one become available in the future.
>
> Thank you again for your time in reviewing our manuscript!

---

> > ### Comment · Reviewer_bhmA · 2024-08-16
> > **Follow-up Response**
> >
> > I appreciate authors' effort in rebuttal. My concern has been addressed so I will keep my rating.

---

### Official Review · Reviewer_BsqT · 2024-07-23
**A Multi-Task Benchmark Dataset?**

**Rating:** 4
**Confidence:** 2

**Review:**

See other areas for pros.

Cons:
- The authors argue that NanoBaseLib is a “A Multi-Task Benchmark Dataset”, but in their limitations raise that “benchmarks presented in the manuscript are for illustration purposes”. This is contradictory and raises substantial concerns. This is further illustrated by “We will constantly update NanoBaseLib when new public datasets are available in the future. We hope this will mitigate the aforementioned limitations.”. I would expected this manuscript to provide the benchmark dataset, if the title says so. While I understand that benchmarks need to be updated particularly in fields that are developing quickly and are prone to changes, the wording implies that the dataset assembled here cannot at all be used for benchmarking.
- Can the authors comment on the quality of the data assembled? I am specifically interested in the assessment to which level the data can be considered “ground truth”?
- Of the 4 tasks referenced, 2 appear to be solved most commonly by HMMs. Please provide further information why more advanced machine learning may be necessary or useful as this may not be clear. In this context, I would also recommend highlighting tasks in terms of priority or likely benefits such that researchers interested in this topic may pick a task that is most promising.
- Based on Table 5 in the Supplement, it appears that for most tasks the raw signal current is required. This is not provided by the resource directly and thus 3 of 4 tasks cannot be approached. How do the authors anticipate that ML scientists would get access to the raw signal? Unless I overlooked it, it seems that he authors must provide a pipeline to download the raw signal such that any new model can be trained on the Nanopore signal.
- Please provide references for all tools used and benchmarked. Please also fix all links requiring logins, e.g. https://community.nanoporetech.com/downloads
- Please provide explicit links to all raw data repositories. E.g. SRP363295 cannot be found by simply googling for the accession and thus requires some in-depth knowledge about the relevant repositories.

**Strengths:**

The paper provides a good introduction for non-experts in the field of Nanopore sequencing. Significant amounts of the paper are dedicated to describing the required knowledge about the different steps necessary to process Nanopore sequencing data. Achieving the stated goal of creating a dataset/framework to attract more ML researchers to the field of nanopore sequencing.

**Additional Feedback:**

-

**Clarity:**

Overall, the description of the underlying technology is clear and the paper well written. Some minor points where clarity can be improved (besides the mentioned issues in Review):
Page 6, line 219, “In the sequel, we will discuss”. I believe saying “In the following, we will discuss” matches the intended meaning better.
Please align the color schemes of Figure 2 and Supplemental Figure 5.
I believe that results should not be displayed in a Tabular format. A simple barplot provides a much more accessible overview of the results (Table 2&3). If you disagree with this, please ensure consistency and turn Figure 2 into a table as well.

**Correctness:**

Most of the used metrics appear to be applied correctly. Using the polyA detection rate to benchmark doesn’t appear useful to me. As stated previously in the paper, filtering out sequences without polyA tail is a core aspect of the data quality assurance step of polyA detection. I believe the secondary metric used accuracy of polyA tail length estimation appears much more meaningful. I would therefore suggest switching Figure 2 & Supplemental Figure 5.
In Supplementary Table 4, there are only two and three samples for the two eGFP datasets, respectively. According to the “Ground truth acquisition” section in 4.2, I would expect six samples each, similar to the “ont_polya_standard” dataset.
Rodan seems to be missing in Supplemental Table 7. Can this tool not be configured?

**Documentation:**

The dataset overview is well structured, and the different steps for which the various benchmarking datasets can be used are well described. The only thing missing regarding dataset documentation would be an overview of the different locations where the raw (unprocessed) data was acquired.
More emphasis can be spent on documenting the different tools used in this study. Adding an additional supplemental table to provide an overview of all the tools used would help a lot in understanding tool limitations. It is, for example, not immediately apparent why Rodan was not included in Table 2. When checking Supplemental Table 8, it becomes apparent that it is only usable for RNA. I also assume that all tools supporting m5C and m6A were used for the respective datasets. It would be great if I could easily confirm this in a table stating tool limitations.
Additionally, Supplemental Figure 4 should be updated to provide a complete overview of all the tools used in this study.

**Ethics:**

The paper assembles publicly available data. Please add (if available) the licences under which each dataset is available.

**Limitations:**

The authors briefly address the limitations of the paper in the appendix. They should be more prominently displayed as part of the main paper - and addressed before publication.

**Opportunities For Improvement:**

See cons in review.

**Relation To Prior Work:**

The authors briefly state how their work relates to prior publications. I don’t have enough experience in the field if other datasets/publications are missing.

**Summary And Contributions:**

The paper describes a collection of nanopore sequencing benchmarking datasets and a framework to run benchmarks for several tools covering the entire Nanopore sequencing data analysis pipeline. The described accessibility of the dataset combined with comprehensive descriptions of processing steps should enable ML scientists with limited experience in the field of Nanopore sequencing to easily start developing novel machine learning models to improve the various aspects of Nanopore sequencing data analysis.

---

> ### Author Response · Authors · 2024-08-16
> **Rebuttal by Authors Part 3**
>
> Dear Reviewer #BsqT:
>
> This is the third part of our response.
>
> > Please provide references for all tools used and benchmarked. Please also fix all links requiring logins, e.g. https://community.nanoporetech.com/downloads
>
> > More emphasis can be spent on documenting the different tools used in this study. Adding an additional supplemental table to provide an overview of all the tools used would help a lot in understanding tool limitations.
> > Additionally, Supplemental Figure 4 should be updated to provide a complete overview of all the tools used in this study.
>
> > It would be great if I could easily confirm this in a table stating tool limitations.
>
> Thanks very much for your valuable suggestions! We will respond to all your concerns about the tools here.
>
> Your suggestion to add a new tools overview and limitations table is excellent! In response, **we added a new table titled “Processing tools overview and limitations” on our website https://nanobaselib.github.io/pipeline.html** (A screenshot is shown in Fig. 3 of the attached pdf. The pdf is located at the end of the first part). This table summarized the download links and limitations of all tools. We will update Appendix Table 6 accordingly in the revised version.
>
> Regarding the references for the tools used and benchmarked, we have included them in our manuscript. For example, Citation 26 is provided for minimap2 in Line 184, and Citations 35-39 are provided for the modification detection benchmark in Lines 307-308. In the revised version, we will also add these references to Appendix Table 6 to make the information even clearer.
>
> Regarding the login access, these are generally related with tools (Guppy, Dorado) developed by Oxford Nanopore Technologies (ONT) company, which produces the Nanopore sequencing devices. Since ONT is a commercial company, users need to register an account (for free) on their website to download the software.
>
> > Rodan seems to be missing in Supplemental Table 7. Can this tool not be configured?
>
> We selected two types of benchmark tools for base calling: official ONT basecallers (e.g., Guppy, Bonito, and Dorado) and third-party basecallers (e.g., Causalcall and Rodan). Supplemental Table 7 only summarized the configurations of the official ONT basecallers. The third-party basecallers do not have unified configurations that is why Rodan is missing from this table. Users can configure them according to the documentation provided by each tool.
>
> > I also assume that all tools supporting m5C and m6A were used for the respective datasets.
>
> In modification detection task, some tools can be used for detecting all types of modifications (e.g., Tombo *de novo*), while others are specialized for specific types (e.g., CHEUI is suitable for m6A and m5C). Additionally, some tools are designed to detect only one type of modification (e.g., m6Anet only for m6A). In the newly added table on our website https://nanobaselib.github.io/pipeline.html (A screenshot is shown in Fig. 3 of the attached pdf. The pdf is located at the end of the first part), we have listed the limitations of all tools.
>
> > Using the polyA detection rate to benchmark doesn’t appear useful to me. As stated previously in the paper, filtering out sequences without polyA tail is a core aspect of the data quality assurance step of polyA detection. I believe the secondary metric used accuracy of polyA tail length estimation appears much more meaningful. I would therefore suggest switching Figure 2 & Supplemental Figure 5.
>
> Thank you for your suggestions. We will switch Figure 2 and Supplemental Figure 5 in the revised version.
>
> >  In Supplementary Table 4, there are only two and three samples for the two eGFP datasets, respectively. According to the “Ground truth acquisition” section in 4.2, I would expect six samples each, similar to the “ont_polya_standard” dataset.
>
> The two datasets have different naming conventions, but both of them have six samples or subsamples corresponding to the ground truth labels. The “ont_polya_standard” raw dataset is organized by polyA length, resulting in six samples (10x, 15x, 30x, 60x, 80x, 100x). However, the eGFP raw datasets are organized by kit and replicate name. Each eGFP sample also contains six subsamples (10x, 30x, 40x, 60x, 100x, 150x), which correspond to the polyA tail length labels.
>
> >  I believe that results should not be displayed in a Tabular format. A simple barplot provides a much more accessible overview of the results (Table 2&3). If you disagree with this, please ensure consistency and turn Figure 2 into a table as well.
>
> Thank you for the suggestions. We will convert Figure 2 into a table in the revised version to keep consistency. As a supplement, we have put the figures on our website (https://nanobaselib.github.io/ploya.html).
>
> We hope our responses could clear up your concerns; please let us know if you have any follow-ups or further concerns. Thank you again for taking the time to review our manuscript!

---

> ### Author Response · Authors · 2024-08-16
> **Rebuttal by Authors Part 2**
>
> Dear Reviewer #BsqT:
>
> This is the second part of our response.
>
> > Of the 4 tasks referenced, 2 appear to be solved most commonly by HMMs. Please provide further information why more advanced machine learning may be necessary or useful as this may not be clear. In this context, I would also recommend highlighting tasks in terms of priority or likely benefits such that researchers interested in this topic may pick a task that is most promising.
>
> In NanoBaseLib, the Base Calling (BC), Modification Detection (MD), and PolyA Detection (PD) tasks all include at least one deep learning-based method. (Appendix Table 5 only presents typical models but not all baseline models.) However, the accuracy of these tasks is far from satisfactory. For instance, the average base calling accuracy of Nanopore RNA sequencing is ∼90%, while the aim is ∼99.9% (also mentioned in Abstract Line 5-8 of the manuscript). **These tasks need more powerful deep learning models to improve performance.**
>
> The Segmentation and Event Alignment (SA) task is essentially an unsupervised task. Therefore, the commonly adopted methods for this task are generally based on HMMs. The huge challenge in this task is lacking the high quality labels. **We think unsupervised learning or semi-supervised learning will play an essential role in this task, underscoring the urgent need for contributions from the ML community.**
>
> Regarding the importance of the four tasks, it is really difficult to rank them and we think we should not make recommendations. Each task is important for different biological applications. BC and SA tasks are the fundamental cornerstones of Nanopore sequencing data analysis. If one could make significant improvements in these tasks, e.g. bring the base calling accuracy to 99.9%, it will have a significant impact on the Nanopore sequencing community. The MD task is important for a new field called epi-transcriptomics that is becoming popular in the past decade [3].  PD is associated with basecalling, which is important for sequencing quality control. Application-wise, it is important to study some biological functions of viruses [4]. In summary, we think all tasks are important. However, we have added more detailed information about each model for the four tasks on our website (https://nanobaselib.github.io). Researchers can explore and select the task that best suits their interests.
>
> Reference:
>
> [3] Wiener D, Schwartz S. The epitranscriptome beyond m6A[J]. Nature Reviews Genetics, 2021, 22(2): 119-131.
>
> [4] Brouze A, Krawczyk P S, Dziembowski A, et al. Measuring the tail: Methods for poly (A) tail profiling[J]. Wiley Interdisciplinary Reviews: RNA, 2023, 14(1): e1737.
>
> > How do the authors anticipate that ML scientists would get access to the raw signal?
>
> > Please provide explicit links to all raw data repositories. E.g. SRP363295 cannot be found by simply googling for the accession and thus requires some in-depth knowledge about the relevant repositories.
>
> >  The only thing missing regarding dataset documentation would be an overview of the different locations where the raw (unprocessed) data was acquired.
>
> Thank you for highlighting the need to improve the dataset documentation in the Appendix. We will address all your concerns regarding raw data availability and license here.
>
> **In addition to the documentation provided in the Appendix of the manuscript, we have launched a website (https://nanobaselib.github.io) that offers more detailed information and resources.** On the dataset page (https://nanobaselib.github.io/dataset.html), you can find comprehensive details about the datasets, including download links, accession numbers, meta information, and more. Both the raw dataset and our preprocessed data are available for download. **To further enhance accessibility, we added a new page (https://nanobaselib.github.io/raw.html) to simplify the process of downloading raw data.** Screenshots of the website are provided in Figures 1-2 in the attached pdf (end of the first part). We will also clarify the download links in the Appendix in the revised version of the manuscript.
>
> The accession number "SRP363295" serves as a unique identifier for the raw dataset, directing users to the original repositories hosted by NCBI or EBI database. Through this number, users can access associated information and download the raw dataset. Accession number starting with “PRJEB” should be downloaded from EBI-ENA (https://www.ebi.ac.uk/ena/browser/home). Accession number starting with “SRP” should be downloaded from NCBI-SRA (https://www.ncbi.nlm.nih.gov/sra). So SRP363295 can be found at https://www.ncbi.nlm.nih.gov/sra/?term=SRP363295. However, our newly launched webpage (https://nanobaselib.github.io/raw.html) offers a more streamlined and user-friendly interface for navigating and downloading raw data. Regarding the license, all raw data are sourced from publications or the NCBI/EBI database. We did not find the licenses associated with these data.

---

> ### Author Rebuttal · Authors · 2024-08-16
>
> Dear Reviewer #BsqT:
>
> We sincerely appreciate your thorough review and valuable feedback on our paper! We have carefully considered your comments and concerns and provided point-by-point responses below. The responses are divided into three parts, and this is the first part.
>
> > The authors argue that NanoBaseLib is a “A Multi-Task Benchmark Dataset”, but in their limitations raise that “benchmarks presented in the manuscript are for illustration purposes”. I would expected this manuscript to provide the benchmark dataset, if the title says so. While I understand that benchmarks need to be updated particularly in fields that are developing quickly and are prone to changes, the wording implies that the dataset assembled here cannot at all be used for benchmarking.
>
> Thank you for highlighting the need to improve the writing in our manuscript. The misunderstanding mentioned above was indeed a result of unclear writing. We will clarify in the revised version that the baseline results in NanoBaseLib are valid and that the dataset can indeed be used for benchmarking.
>
> The core concept of NanoBaseLib is "**One Dataset, Multiple Tasks**," which allows for the analysis of various tasks on a single dataset. This approach contrasts with traditional methods, where different datasets are processed separately for a single task. When we mentioned that the "benchmarks presented in the manuscript are for illustration purposes," we simply meant that **the experiments included in our paper were benchmarked on only a subset of our 16 datasets, and they aim to illustrate that NanoBaseLib has the capability to process multiple tasks on a single dataset, even though the benchmarks are not exhaustive.** For example, traditional methods typically benchmark the Base Calling (BC) task across several datasets or several species, while in NanoBaseLib, we benchmarked BC on two datasets rather than on all datasets. However, it does not imply that “the datasets assembled here cannot be used for benchmarking”.
>
> Given that we designed four distinct tasks in NanoBaseLib, **benchmarking all datasets on all the baselines of all tasks** would result in an enormous workload, which unfortunately falls outside the scope of this manuscript.  However, our NanoBaseLib has released 16 public datasets, which are sufficient for building more comprehensive benchmarks. Additionally, we developed a new software package (https://github.com/nanobaselib/NanoBaseLib) that streamlines the incorporation of new datasets, enabling efficient preprocessing. Researchers can build more comprehensive benchmark results by running our software package on the other datasets we released in NanoBaseLib. The benchmarks included in our manuscript can now serve as a solid foundation for more comprehensive benchmarking in the future, even though they are not exhaustive.
>
> We will also clarify this in the Limitations section to clear up the misunderstandings.
>
> Regarding dataset updates, as mentioned above, building more comprehensive benchmarks across additional datasets is one of the future goals of NanoBaseLib. Additionally, new Nanopore datasets will continue to emerge, highlighting the necessity of updating the datasets. Therefore, we designate the current version (including baselines and datasets) as *NanoBaseLib v1.0*.
>
> > Can the authors comment on the quality of the data assembled? I am specifically interested in the assessment to which level the data can be considered “ground truth”?
>
> By following a unified preprocessing pipeline, we processed all data starting from the raw current signals of Nanopore sequencing. The use of standardized software configurations, a consistent processing pipeline, a unified storage format, our newly developed software package (https://github.com/nanobaselib/NanoBaseLib), and comprehensive documentation (https://nanobaselib.github.io) collectively ensures the high quality of the data we have assembled. We believe these measures significantly enhance the reliability and reproducibility of our dataset, making it a robust resource for further research and analysis.
>
> Regarding the ground truth, we have tried our best to obtain accurate data for each task by following the standard practices in the Nanopore sequencing community, which were utilized in papers published in top venues such as *Nature Method*, *Nature Biotechnology*, etc. For certain tasks, such as modification detection and polyA detection, the ground truth data are derived from biological experiments [1, 2]. We are confident that the obtained ground truths will be accepted by the bioinformatics community and can be used for future benchmark studies.
>
> Reference:
>
> [1] Hendra C, Pratanwanich P N, Wan Y K, et al. Detection of m6A from direct RNA sequencing using a multiple instance learning framework[J]. Nature methods, 2022, 19(12): 1590-1598.
>
> [2] Krause M, Niazi A M, Labun K, et al. tailfindr: alignment-free poly (A) length measurement for Oxford Nanopore RNA and DNA sequencing[J]. RNA, 2019, 25(10): 1229-1241.
>
> > The authors briefly address the limitations of the paper in the appendix. They should be more prominently displayed as part of the main paper - and addressed before publication.
>
> Thank you for your valuable comments! Due to paper space constraints, we placed the limitation discussion in the Appendix. Following the suggestions of Reviewer #KfCJ, we will shorten the introduction section and move the limitation section to the main paper.
>
> >  Page 6, line 219, “In the sequel, we will discuss”. I believe saying “In the following, we will discuss” matches the intended meaning better
>
> Thanks for pointing it out and we will revise it in the next version.
>
> > Please align the color schemes of Figure 2 and Supplemental Figure 5.
>
> Thank you for your suggestions. In response, we updated it on our website https://nanobaselib.github.io/ploya.html (screenshot shown in attached pdf Figure 4). We will also update the figures in the revised version.

---

### Official Review · Reviewer_k9pY · 2024-07-24
**Useful introduction of nanopore data and techniques to the ML community**

**Rating:** 9
**Confidence:** 4
**Clarity:** yes

**Review:**

The paper is well reasoned and written. It describes nanopore sequencing data and challenges at a level that is understandable to those new to the field. While there exist several other benchmark datasets, as described by the authors in Related Work, this dataset is the most comprehensive and end-to-end to date, which greatly increases its significance. In general, nanopore sequencing is an exciting area of technological development, and the challenges in data analysis are a ripe area for ML. Historically, though, the complexities of the data and how it's processed poses a challenge to those outside of the area.

**Strengths:**

Though not a panacea -- this dataset and benchmark pipeline goes a step toward making this class of data more approachable and standardized. Te relevance to the broader research community may be perceived to be not very significant due to the obscurity of nanopore sequencing technology to many in the ML community. However, this exciting are of development is a great application for those interested in developing new ML techniques to solve new and important challenges.

**Additional Feedback:**

-The paper states in a couple of locations that Illumina sequencing is limited to read lengths <150. This is not accurate, some Illumina kits support read lengths up to 300.
-The paper claims library prep for nanopore sequencing is much simpler than for other methods, however, this depends greatly on the prep kit used.
-Can the authors make sure that their claim of 90% sequencing accuracy is considering the latest pore chemistry that ONT has released (e.g. R10) which claims to have a much higher accuracy?

**Correctness:**

Did the authors make sure to check if any of the analysis tools they benchmarked were trained on any of their datasets?

**Documentation:**

yes

**Ethics:**

no issues

**Limitations:**

There may be some upper limit on the accuracy that can be achieved due to the fact that nanopore sequencing directly senses the nucleic acid strand of a single-molecule. This means that true ground truth may be impossible to achieve due to the presence of unknown epigenetic of damaged bases that affect the current signal levels.

**Opportunities For Improvement:**

It could be useful to describe some of the ML approaches used in the tools that are benchmarked in this paper. This could help ML researchers understand the current approaches and SOTA, further enabling new comers in the field.

**Relation To Prior Work:**

yes

**Summary And Contributions:**

This submission introduces a nanopore DNA/RNA sequencing benchmark dataset as a means to make this area of technology more accessible to those who want to develop new ML tools to address outstanding challenges in nanopore signal analysis. The authors do this by collecting and standardizing a number of publicly available DNA/RNA nanopore sequencing datasets, then develop a pipeline for benchmarking different analysis tasks on these datasets. The authors set benchmarks for the most commonly used tools that are currently available.

---

> ### Author Rebuttal · Authors · 2024-08-16
>
> Dear Reviewer #k9pY:
>
> Thank you for your thoughtful review of our work! We hope to address your comments and concerns point-by-point:
>
> > It could be useful to describe some of the ML approaches used in the tools that are benchmarked in this paper. This could help ML researchers understand the current approaches and SOTA, further enabling new comers in the field.
>
> Thank you for your constructive suggestions! In response, we have updated our NanoBaseLib website to provide more details about the baselines in four tasks. For example, we have illustrated the typical model architecture in the base calling task (https://nanobaselib.github.io/basecall.html). Baseline descriptions of other tasks are also provided on our website (please visit the other task pages on our website: https://nanobaselib.github.io). We will also clarify the details in the corresponding subsections of Section 4 and the Appendix in the revised version.
>
> > There may be some upper limit on the accuracy that can be achieved due to the fact that nanopore sequencing directly senses the nucleic acid strand of a single-molecule. This means that true ground truth may be impossible to achieve due to the presence of unknown epigenetic of damaged bases that affect the current signal levels.
>
> We appreciate your insightful comments and we will add this to our limitation section!
>
> > Did the authors make sure to check if any of the analysis tools they benchmarked were trained on any of their datasets?
>
> We have carefully checked all baseline tools and confirmed that none of them were trained on NanoBaseLib benchmarking datasets. We will also clarify this in Section 4.
>
> > The paper states in a couple of locations that Illumina sequencing is limited to read lengths <150. This is not accurate, some Illumina kits support read lengths up to 300.
>
> Thank you for pointing out this! We will update the 150bp to **300bp** in Lines 22 and 38 and add the following reference.
>
> Reference:
>
> [1] Hu T, Chitnis N, Monos D, et al. Next-generation sequencing technologies: An overview[J]. Human Immunology, 2021, 82(11): 801-811.
>
> > The paper claims library prep for nanopore sequencing is much simpler than for other methods, however, this depends greatly on the prep kit used.
>
> Thanks for pointing this out! We will update the following sentence in Line 40
>
> “Second, the sequencing library preparation is much simpler than second-generation sequencing”
>
> to
>
> “Second, the sequencing library preparation is **generally**  simpler than second-generation sequencing”.
>
> > Can the authors make sure that their claim of 90% sequencing accuracy is considering the latest pore chemistry that ONT has released (e.g. R10) which claims to have a much higher accuracy?
>
> 90% is the average sequencing accuracy on R9.4 for RNA base calling, without considering the latest R10 pore since there is currently a lack of extensive publicly available data of this version. We will include the pore version information in the revised manuscript and add the following sentence to the discussion:
>
> “Note that all public datasets we collected by the submission are based on R9.4 Nanopore chemistry. The performance on the four tasks of R10 Nanopore chemistry remains to be further investigated.”
>
> Thank you again for taking the time to review our manuscript!

---

> > ### Comment · Reviewer_k9pY · 2024-08-18
> > **follow up response**
> >
> > I appreciate the responses to my concerns and will keep my rating.

---

### Official Review · Reviewer_KfCJ · 2024-07-25

**Rating:** 6
**Confidence:** 2
**Correctness:** The claims are correct.

**Review:**

Quality: The technical quality of the dataset construction is good. The authors have collected raw signal data from online databases and implemented a unified preprocessing pipeline, ensuring data consistency across various tasks.

Clarity: The clarify can be improved. Critical details such as batch effect handling and comprehensive dataset statistics are insufficiently addressed.

Originality: While the dataset utilizes publicly available data, the implementation of a unified preprocessing pipeline represents a methodological contribution in reducing bias across datasets.

Pros:
1.	This work proposes a solid dataset construction pipeline.
2.	The dataset website provides sufficient information and is well-designed.
3.	The proposed dataset and benchmark will shed light on future research in this direction.

Cons:
1.	There should be more discussion on batch effect handling.
2.	The provided dataset statistics are insufficient.
3.	Writing can be improved.

**Strengths:**

1.	The development of NanoBaseLib represents a substantial technical achievement in curating a multi-task benchmark dataset for nuclear acids sequencing.

2.	By focusing on raw signal data and implementing a unified preprocessing pipeline, the authors ensure data consistency and mitigate biases that may arise from varying preprocessing methods.

**Additional Feedback:**

N.A.

**Clarity:**

The clarity can be improved from the following perspectives:
1.	Structure Improvement: The overall structure could benefit from revision. The introduction is overly detailed. Streamlining this section to focus on the essential motivations and contributions of NanoBaseLib would enhance clarity and engagement.
2.	Missing Dataset Details: Crucial details such as dataset size and comprehensive statistics are absent. Including these information is essential for readers to assess the dataset's scope, variability across tasks, and applicability to different machine learning models.

**Documentation:**

The details are not sufficient. The statistics of the dataset is missing, such as the number of sequences, mean length, etc. The dataset website is well structured and easy to use.

**Ethics:**

There are no ethical concerns.

**Limitations:**

The authors addressed the limitations in the appendix, but potential negative societal impact is not discussed in the paper.

**Opportunities For Improvement:**

1.	The writing can be improved to clearly present the motivation and method.
2.	Critical details such as batch effect handling and comprehensive dataset statistics should be addressed.

**Relation To Prior Work:**

The relation to prior work is clearly discussed.

**Summary And Contributions:**

The submission introduces NanoBaseLib, a novel multi-task benchmark dataset tailored for evaluating machine learning models in nanopore sequencing. NanoBaseLib encompasses diverse tasks such as basecalling, PolyA detection, segmentation and event alignment, and modification detection. To create the dataset, the authors first collected data from NCBI and EBI. The authors collected raw signal data instead of the preprocessed data since different datasets uses different preprocess methods, which may introduce unwanted bias. The authors designed a unified pre-processing pipeline to standardize the handling of raw signal data. They also analysed test datasets with various baseline methods for four benchmark tasks, and developed a software package to easily access these results.

---

> ### Author Rebuttal · Authors · 2024-08-16
>
> Dear Reviewer #KfCJ:
>
> Thank you for your time and constructive comments! We hope to address your comments and concerns point-by-point:
>
> > The provided dataset statistics are insufficient. / The details are not sufficient. The statistics of the dataset are missing, such as the number of sequences, mean length, etc.
>
> Thank you for highlighting the importance of detailed dataset statistics. **In response, we added a new table titled “Dataset Statistics” on our website (https://nanobaselib.github.io/dataset.html) that includes key metrics such as dataset size, the number of reads, mean raw signal length, and mean base sequence length.** A screenshot of the updated website is shown in Fig. 1 of the attached pdf. We will also add the statistics table to the Appendix in the revised version of the manuscript. Please feel free to suggest any additional dataset information that we should consider adding to the webpage.
>
> > There should be more discussion on batch effect handling.
>
> Thank you for the suggestion! Typically, the “batch effect” refers to systematic technical differences introduced by device, time, and place-dependent experimental conditions [1]. In Nanopore sequencing, the “batch effect” may be introduced by different kits, reagents, and other factors.
>
> One of the important factors causing “batch effect” is the sequencing bias generated by different sequencing devices. Firstly, each ONT sequencing device (e.g. MinION) will generate a set of device-specific parameters to convert the raw signal into pico Ampere (pA) values. This ensures the consistency between different sequencing devices. Subsequently, different downstream tools will standardize or normalize the signal further. Both of these processes are all utilized in our preprocessing pipeline, which essentially remove the batch effects to a certain extent. **In response, we added a new webpage explaining more details about the raw signal conversion and different normalization process on our website (https://nanobaselib.github.io/signal.html).** A screenshot of the new page is shown in Fig. 2 of the attached pdf.
>
> We also plan to add the following sentences to the discussion in the revised version:
>
> “Each fast5 file is accompanied by a set of device-specific parameters, which are used to convert the raw signal into picoampere (pA) values. Different downstream tools further standardize or normalize the signal. These processes help to remove batch effects between different datasets to a certain extent. The signal conversion and normalization processes in our preprocessing pipeline are detailed in the Appendix.”
>
> We are happy to provide further responses if you have more questions about batch effect handling.
>
> Reference:
>
> [1] Lazar C, Meganck S, Taminau J, et al. Batch effect removal methods for microarray gene expression data integration: a survey[J]. Briefings in bioinformatics, 2013, 14(4): 469-490.
>
> > The authors addressed the limitations in the appendix, but potential negative societal impact is not discussed in the paper.
>
> Thank you for your comments. All raw datasets were collected from public resources, and we preprocessed them using a standard pipeline. There do not appear to be any privacy issues. However, the datasets could still be exploited for unforeseen purposes, such as commercialization—where anonymized genetic data might be used by companies to develop targeted products, potentially compromising privacy or leading to social inequities, like expensive drugs limited to specific populations. Additionally, the data might be repurposed for unintended research, possibly contradicting the original data providers' intent or causing societal harm. We will add this discussion to the limitations section in the revised version.
>
> >  The writing can be improved to clearly present the motivation and method. / Structure Improvement: The overall structure could benefit from revision. The introduction is overly detailed. Streamlining this section to focus on the essential motivations and contributions of NanoBaseLib would enhance clarity and engagement.
>
> Thank you for your feedback on the organization and writing of our paper. To accommodate readers with diverse backgrounds, we added extensive details to the Introduction to ensure that those unfamiliar with Nanopore Sequencing can better engage with the content. In the revised version, we will incorporate your suggestions and refine the writing to make the Introduction more concise, focusing on the motivation, methods, and key contributions.
>
> We plan to move the section “Pros and Cons of Nanopore” to the Appendix and summarize it with one sentence:
>
> “Compared with Illumina sequencing, Nanopore sequencing preserves the epi-transcriptomic information on DNA/RNA molecules but suffers from lower base-calling accuracy (further discussion in the Appendix).”
>
> Furthermore, we will restructure the last paragraph “Our solution” in Introduction to clarify our contributions. In this work, we present NanoBaseLib, a multi-task benchmark dataset. Our main contributions are as follows:
>
> - We build a comprehensive Nanopore sequencing dataset by integrating 16 public datasets with over 30 million reads, all preprocessed using our proposed unified pipeline.
>
> - We benchmark four key tasks of Nanopore sequencing based on the unified, preprocessed dataset. These tasks include base calling, polyA detection, segmentation and event alignment, and RNA modification detection (m6A, m5C, hm5C, inosine, pseudouridine).
>
> - We develop a software package that streamlines the incorporation of new datasets, enabling efficient preprocessing and integration of newly available Nanopore data.
>
> - We deploy a user-friendly website to facilitate easier navigation of datasets, tasks, and benchmark results.
>
> We hope our responses could clear up your concerns; please let us know if you have any follow-ups or further concerns. Thank you again for taking the time to review our manuscript!

---

> > ### Comment · Reviewer_KfCJ · 2024-08-22
> >
> > Thank authors for the rebuttal. The rebuttal addressed most of my questions so I increased my rating to borderline accept.

---

### Decision · Program_Chairs · 2024-09-26

**Decision:**

Accept (Poster)

**Comment:**

The article presents NanoBaseLib, a multi-task benchmark dataset designed for nanopore sequencing, aimed at improving the accuracy of nanopore sequencing data analysis. The dataset focuses on four critical tasks in nanopore data analysis: Base Calling (BC), PolyA Detection (PD), Segmentation and Event Alignment (SA), and Modification Detection (MD). NanoBaseLib integrates 16 public datasets, totaling over 30 million reads, and has been preprocessed using a uniform workflow. The article describes how to evaluate the performance of different algorithms, including the selection of baseline models and the calculation of evaluation metrics. By comparing the performance of various models, NanoBaseLib provides a reliable reference framework for researchers.
This paper was somewhat difficult to evaluate because the reviewer scores ranged from 4 - 9. However, when reading more carefully, the primary concern of the reviewer scoring 4 was the quality of the data, which the authors addressed during the rebuttal period by clarifying the language.
Overall, the paper has been substantially enhanced based on the reviewers' feedback, and the authors have effectively addressed the concerns raised.